# NCF4 attenuates colorectal cancer progression by modulating inflammasome activation and immune surveillance

Longjun Li[1,9], Rudi Mao[1,9], Shenli Yuan[2,9], Qingqing Xie[1], Jinyu Meng[1], Yu Gu[3], Siyu Tan[4], Xiaoqing Xu[5], Chengjiang Gao[4], Hongbin Liu[6], Chunhong Ma[4], Si Ming Man[7,10]✉, Xiangbo Meng[1,10]✉, Tao Xu[1,10]✉ & Xiaopeng Qi[1,8,10]✉

The spatiotemporal regulation of inflammasome activation remains unclear. To examine the mechanism underlying the assembly and regulation of the inflammasome response, here we perform an immunoprecipitation-mass spectrometry analysis of apoptosis-associated speck-like protein containing a CARD (ASC) and identify NCF4/1/2 as ASC-binding proteins. Reduced *NCF4* expression is associated with colorectal cancer development and decreased five-year survival rate in patients with colorectal cancer. NCF4 cooperates with NCF1 and NCF2 to promote NLRP3 and AIM2 inflammasome activation. Mechanistically, NCF4 phosphorylation and puncta distribution switches from the NADPH complex to the perinuclear region, mediating ASC oligomerization, speck formation and inflammasome activation. NCF4 functions as a sensor of ROS levels, to establish a balance between ROS production and inflammasome activation. NCF4 deficiency causes severe colorectal cancer in mice, increases transit-amplifying and precancerous cells, reduces the frequency and activation of CD8⁺ T and NK cells, and impairs the inflammasome-IL-18-IFN-γ axis during the early phase of colorectal tumorigenesis. Our study implicates NCF4 in determining the spatial positioning of inflammasome assembly and contributing to inflammasome-mediated anti-tumor responses.

The canonical inflammasome is an important component of the innate immune system and is assembled as a cytosolic multiprotein complex by an inflammasome sensor protein, apoptosis-associated speck-like protein containing a CARD (ASC), and caspase-1 in response to pathogenic and sterile insults[1,2]. ASC speck formation and inflammasome assembly are essential for caspase-1 maturation and the subsequent cleavage of proinflammatory cytokines and Gasdermin D (GSDMD)[3,4]. The multilayered regulatory processes and the mechanisms underlying inflammasome activation have been extensively studied, and the fundamental roles of inflammasomes under both pathological and physiological conditions have also been described[5–8]. However, the spatiotemporal regulation of inflammasome assembly

and the downstream signaling events that trigger the initiation of the formation of this signaling hub remain unclear.

NLRP3 is the most extensively studied inflammasome due to its extensive interactions with different signaling pathways and the roles it plays in cancer and infectious and inflammatory diseases[8]. NLRP3 was initially reported to localize to the endoplasmic reticulum (ER) and mitochondria close to the perinuclear region upon inflammasome activation[9]. The adaptor proteins MAVS and STING, which are involved in cytosolic nucleic acid-triggered IFN-I signaling pathways, have been demonstrated to promote NLRP3 inflammasome activation by mediating the recruitment of NLRP3 to mitochondria and the ER, respectively[10,11]. In particular, NLRP3 stimuli can cause trans-Golgi

network (TGN) disassembly, or endosomal accumulation of PI4P and disruption of endosome-to-*trans*-Golgi network trafficking (ETT), which induces endosomal recruitment of NLRP3 and the subsequent inflammasome assembly[12,13]. In addition, the cytoskeleton, the nucleus, stress granules, and other cellular compartments can function as platforms for the trafficking of cellular components and inflammasome assembly[14,15]. Inflammasome components are associated with virtually all organelles within mammalian cells, and the activated NLRP3 inflammasome is thought to sense the disturbance of cellular homeostasis rather than a particular motif within a specific activator[16]. How diverse stimuli converge at a single enzymatic signaling hub across different cellular localizations remains unknown.

The intracellular localization of NLRP3 is necessary to mediate a rapid response to multiple cellular danger signals, such as $K^+$ efflux, $Ca^{2+}$ mobilization, and reactive oxygen species (ROS) production, which have been proposed to drive NLRP3 inflammasome activation[17]. ROS have been speculated to be central and shared common signaling molecules, with their production triggered by a wide range of stimuli to mediate NLRP3 inflammasome activation by regulating both priming and activation signals[18–21]. Electron transport in mitochondria and NADPH oxidase (NOX) in the membrane are two major drivers of ROS production, and thioredoxin-interacting protein (TXNIP) and oxidized mitochondrial DNA (mtDNA) have been reported to mediate mitochondrial ROS-triggered NLRP3 inflammasome activation[22,23]. NADPH oxidase in phagocytes is an evolutionarily conserved antimicrobial complex and the main ROS-producing enzyme during inflammation; it is composed of the membrane-bound catalytic core proteins p22phox and NOX2 (gp91phox) and the cytoplasmic subunits NCF4 (p40phox), NCF1 (p47phox), and NCF2 (p67phox)[24]. However, the function of NADPH oxidase components in the regulation of NLRP3 inflammasome activation remains unclear, and whether ROS derived from NADPH oxidase and mitochondria function as triggers or effectors of the inflammasome activation is a topic of long-standing debate[25–29].

Here, we show that NCF4 contributes to inflammasome activation by interacting with ASC. Notably, the stimulation-induced puncta distribution of NCF4 promotes ASC oligomerization and speck formation. NCF4 deficiency causes severe colorectal cancer in mice treated with the DNA-damage agent azoxymethane (AOM) and inflammatory agent dextran sodium sulfate (DSS). Our work demonstrates that NCF4 positions the inflammasome assembly within the cell and generates ROS that drives inflammasome activation. This NCF4-coordinated response is necessary to trigger inflammasome-mediated activation and increase the population of anti-tumor CD8+ T and NK cells, preventing colorectal tumorigenesis.

## Results

### NCF4 is an ASC-binding protein and associated with colorectal cancer development

To explore the spatiotemporal regulation of ASC speck formation and inflammasome activation, we infected primary WT and $Asc^{-/-}$ bone marrow-derived macrophages (BMDMs) with the bacterium *Francisella novicida* to induce AIM2 inflammasome activation and then performed ASC IP-MS to identify proteins that interacted with ASC (Fig. 1a). We compared the IP products between WT BMDMs and $Asc^{-/-}$ BMDMs, and found that the components of the NADPH oxidase complex NCF4, NCF1, and NCF2 interacted with ASC. Furthermore, NCF4 showed a much greater specificity for ASC than either NCF1 or NCF2 (Fig. 1b and Supplementary Data 1). An analysis of NCF4, NCF1, and NCF2 homology revealed that the PX domain was shared among all three proteins, and the PB1 domain was carried by NCF4 and NCF2 (Supplementary Fig. 1a). The interaction between ASC and NCF4 was confirmed by co-IP analysis in HEK293T cells transfected with ASC and Flag-tagged full-length NCF4 and NCF4 fragments (Fig. 1c). Notably, the PX and PB1 domains, but not the SH3 domain, were required for NCF4 and ASC interaction (Fig. 1c). The PYD domain but not the CARD

domain of ASC interacted with NCF4 (Fig. 1d). ASC-NCF1 and ASC-NCF2 interactions were also confirmed in HEK293T cells by co-IP analysis. NCF1 interacted with the PYD domain of ASC, and NCF2 interacted with both the PYD and CARD domains of ASC (Supplementary Fig. 1b, c).

We next examined the expression and activation of these three proteins in BMDMs with or without stimulation. Interestingly, the phosphorylation of NCF4 and NCF1, but not that of NCF2, was markedly increased in the presence of LPS, ATP or a combination of LPS and ATP (Supplementary Fig. 1d). To determine the endogenous interaction between ASC and NCF proteins, we treated WT and $Asc^{-/-}$ BMDMs with NLRP3-activating stimuli (LPS and ATP) with short time to avoid formation of insoluble ASC aggregation[30], and performed anti-ASC IP for western blot analysis. Our results revealed that the ASC-interacting complex included NCF1, NCF2, and NCF4 in NLRP3-activated WT BMDMs (Fig. 1e).

NCF4 is a susceptibility gene that is significantly associated with Crohn's disease and colorectal cancer, although the details of the mechanism remain unknown[31,32]. We performed bioinformatic analysis to assess the gene expression of *NCF4*, *NCF1*, and *NCF2* in colon tumors and control tissues from 480 colorectal cancer (CRC) patients and 41 paired tumors and associated normal tissues available in the TCGA database. Interestingly, the expression of *NCF4* and *NCF1*, but not that of *NCF2*, was significantly downregulated in tumor tissues compared to that in control tissues (Fig. 1f). Furthermore, a longer five-year survival rate for colorectal cancer patients correlated with a higher expression of *NCF4*, but not *NCF1* or *NCF2* (Fig. 1g and Supplementary Fig. 1e). Thus, these data indicate that the NADPH oxidase components NCF4, NCF1, and NCF2 form a complex with ASC during inflammasome activation, and these three proteins may play distinct roles in ROS production and inflammasome activation.

### NCF4 contributes to both NLRP3 and AIM2 inflammasome activation

To evaluate the requirement of NCF1, NCF2, and NCF4 in inflammasome activation, we transfected WT BMDMs with gene-specific siRNAs to knockdown *Ncf1*, *Ncf2*, and *Ncf4* gene expression and then treated these BMDMs with different inflammasome activators (Supplementary Fig. 2a). Notably, the levels of activated caspase-1 and released IL-1β in response to LPS and ATP, which activate the NLRP3 inflammasome; dsDNA transfection and infection with *F. novicida*, both as activators of the AIM2 inflammasome, were reduced following transfection with *Ncf1*, *Ncf2*, or *Ncf4* siRNAs. This reduction was most prominent in *Ncf4* siRNA-transfected BMDMs (Supplementary Fig. 2b, c, e). In contrast, caspase-1 activation and IL-1β release in response to *Salmonella enterica* Typhimurium (*Salmonella*) infection, which activates the NLRC4 inflammasome, were not inhibited in cells treated with siRNA targeting *Ncf1*, *Ncf2*, or *Ncf4* genes (Supplementary Fig. 2d, e).

To provide genetic evidence for the role of NCF4 in inflammasome activation, we generated $Ncf4^{-/-}$ mice (Supplementary Fig. 2f, g), which were viable with characteristics similar to those in WT mice under normal conditions, consistent with a previous study[33]. Importantly, caspase-1 activation and IL-1β release in response to NLRP3 and AIM2 activation, but not NLRC4 activation, were substantially reduced in $Ncf4^{-/-}$ BMDMs compared to WT BMDMs (Fig. 2a–d and Supplementary Fig. 2h). Furthermore, ASC speck formation in response to NLRP3 and AIM2 activators was inhibited in the absence of NCF4 (Fig. 2e). To investigate the potential redundant roles of NCF1, NCF2, and NCF4 in inflammasome activation, we transfected WT and $Ncf4^{-/-}$ BMDMs with siRNAs against the *Ncf1* and *Ncf2* genes (Fig. 2f). Caspase-1 activation and IL-1β release in response to NLRP3 and AIM2 activation were further attenuated in $Ncf4^{-/-}$ BMDMs transfected with siRNAs against both *Ncf1* and *Ncf2* (Fig. 2g, h). Overall, these results indicate that NCF4, with contributions from NCF1 and NCF2, plays critical roles in driving the activation of both NLRP3 and AIM2 inflammasomes.

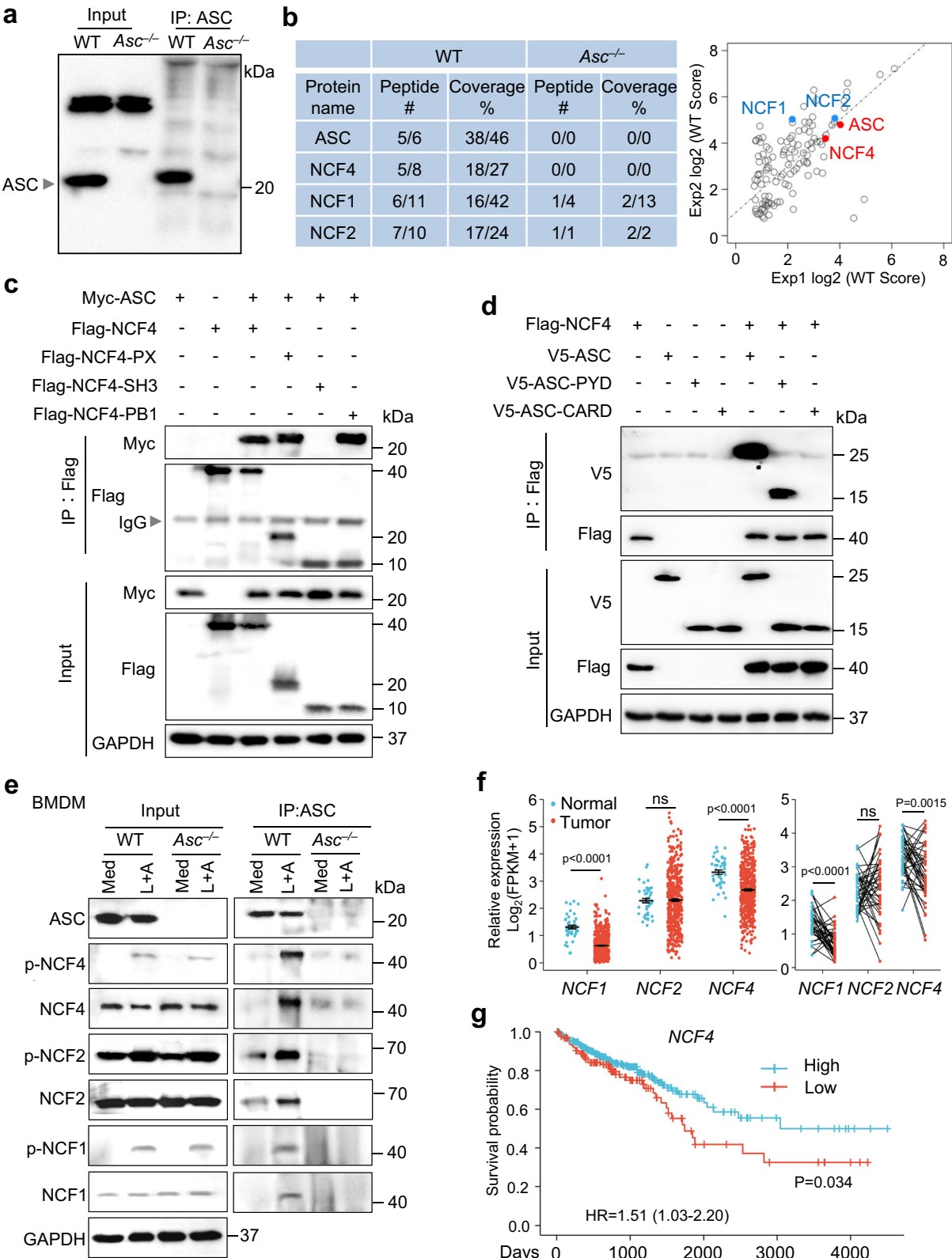

## NCF4 attenuates colorectal cancer progression via inflammasome activation

To investigate the role of NCF4 in colorectal cancer, we intraperitoneally injected littermate WT, *Ncf4*[+/-], and *Ncf4*[-/-] mice with AOM on Day 0 followed by three rounds of DSS treatment (2%) in the drinking water, with assessment of the tumors in the colons on Day 70 (Fig. 3a). *Ncf4*[-/-] mice exhibited the most severe body weight loss after the first round of DSS treatment compared to WT and *Ncf4*[+/-] mice (Fig. 3a). Only 45.0% of the *Ncf4*[-/-] mice survived beyond Day 60, whereas 85.7% of the *Ncf4*[+/-] mice and 91.7% of WT mice survived (Fig. 3b). We found that *Ncf4*[-/-] mice developed a significantly higher number of and larger tumors than WT and *Ncf4*[+/-] mice (Fig. 3c, d and Supplementary Fig. 3a). In addition, the size and weight of the spleen in *Ncf4*[-/-] mice were much larger than those in WT and *Ncf4*[+/-] mice (Supplementary

**Fig. 1 | ASC-interacting protein NCF4 is associated with colorectal cancer development. a** Immunoblot analysis of ASC from the immunoprecipitated products generated with immunoprecipitation with an ASC antibody from the lysates of WT and *Asc*$^{-/-}$ BMDMs infected with *F. novicida* (100 MOI) for 12 h. **b** Mass spectrometry analysis of the IP product in (**a**). The peptides of ASC, NCF4, NCF1, and NCF2 were detected in the IP product with ASC antibody from WT and *Asc*$^{-/-}$ BMDMs in two repeated experiments. **c** Immunoblot analysis of Myc-ASC co-IP with FLAG-NCF4, FLAG-NCF4-PX, FLAG-NCF4-SH3 and FLAG-NCF4-PB1 from lysates of HEK293T cells transfected with the indicated plasmids. **d** Immunoblot analysis of FLAG-NCF4 co-IP with V5-ASC, V5-ASC-PYD, and V5-ASC-CARD from lysates of HEK293T cells transfected with the indicated plasmids. **e** Co-IP analysis of endogenous ASC interacting with total and phosphorylated NCF4, NCF1, and NCF2 in WT and *Asc*$^{-/-}$ BMDMs treated with the NLRP3 activator LPS plus ATP (LPS, 500 ng/mL for 4 h and ATP, 5 mM for 15 min). **f** Gene expression analysis of *NCF1*, *NCF2*, and *NCF4* in total (left) and paired (right, *n* = 41) colorectal tumors (*n* = 480) and control tissues (*n* = 41) from 480 colorectal cancer (CRC) patients of pooled colon and rectal adenocarcinoma datasets in the TCGA database. **g** Correlation analysis between the gene expression of *NCF4* and survival rate of CRC patients. (High, *n* = 466; Low, *n* = 131) Data are from 2 (**a**, **b**) or representative of 3 independent experiments with similar results (**c**–**e**). Wilcoxon signed rank test for (**f**), Log-rank (Mantel–Cox) test for (**g**), *p-value* is indicated in the graph. Source data are provided as a Source Data file.

Fig. 3b). We observed more frequent high-grade dysplasia, increased severity of the damage, and malignant tumors in *Ncf4*$^{-/-}$ mice than in WT mice (Fig. 3e). In addition, a morbidity and mortality analysis revealed that NCF4 exhibited a gene-dose-dependent response to AOM-DSS treatment, and this effect was characteristic of haploinsufficient expression (Fig. 3a–e). Likewise, *Ncf4*$^{-/-}$ mice exhibited more body weight loss and reduction of colon length compared with WT mice in the DSS-induced colitis model, whereas ROS inhibitors NAC or DPI and APO significantly inhibited the disease progression (Supplementary Fig. 3c, d). ROS inhibitors NAC and DPI were also demonstrated to have a positive effect in attenuating DSS-induced colitis and colitis-associated colorectal cancer in both mice and humans[34,35], indicating that NCF4 might prevent colorectal cancer progression through functions other than ROS alone.

Importantly, we found that caspase-1 activation, IL-18 production, and IL-1β production were significantly lower in *Ncf4*$^{-/-}$ mice than in WT and *Ncf4*$^{+/-}$ mice, whereas the production of inflammasome-independent cytokines IL-6 and TNF was similar between WT, *Ncf4*$^{+/-}$, and *Ncf4*$^{-/-}$ mice (Fig. 3f, g). IL-18 plays a critical role in mediating protection against colorectal cancer pathogenesis[36,37]. These data indicate that impaired inflammasome activation in the *Ncf4*$^{-/-}$ mice contributes to the greater tumorigenesis observed. We further compared differences in inflammasome activation and tumor development in co-housed WT, *Ncf4*$^{-/-}$, and *Asc*$^{-/-}$ mice. To prevent the high mortality observed in the *Ncf4*$^{-/-}$ mice, we treated the mice with a lower concentration of DSS (1.5%) (Supplementary Fig. 3e). Under this condition, both *Ncf4*$^{-/-}$ and *Asc*$^{-/-}$ mice lost more body weight than the WT mice (Supplementary Fig. 3e). WT mice all survived beyond Day 70, whereas the survival rate for *Ncf4*$^{-/-}$ mice was 55.6% and *Asc*$^{-/-}$ mice was 71.4% (Supplementary Fig. 3f). Both *Ncf4*$^{-/-}$ and *Asc*$^{-/-}$ mice presented with larger spleens and more extensive colorectal tumorigenesis than WT mice after AOM-DSS treatment (Supplementary Fig. 3g–i). Furthermore, the production of IL-18 and IL-1β, but not that of IL-6 or TNF, was reduced in *Ncf4*$^{-/-}$ and *Asc*$^{-/-}$ mice compared to the WT counterparts (Supplementary Fig. 3j). Furthermore, in a second independent cancer model, we crossed *Ncf4*$^{-/-}$ mice with *Apc*$^{Min/+}$ mice which spontaneously develop colorectal cancer, and found that colon cancer development and spleen size were significantly increased in the absence of NCF4 (Fig. 3h–j). Taken together, these data demonstrate a pivotal role for NCF4 in the prevention of CRC development by mediating inflammasome activation.

### NCF4 guides the intracellular localization of ASC specks
To define the mechanism by which NCF proteins activate inflammasomes, we analyzed the intracellular localization of NCF1, NCF2, and NCF4 under normal and inflammasome-activating conditions. NCF1, NCF2, and NCF4 were diffused throughout untreated WT BMDMs (Supplementary Fig. 4a). Following LPS stimulation in WT, *Asc*$^{-/-}$, and *Nlrp3*$^{-/-}$*Aim2*$^{-/-}$ BMDMs, all three proteins exhibited plasma membrane-bound puncta distribution (Supplementary Fig. 4b). The percentage of punctate-forming cells was approximately 20%, and the colocalization between NCF1 and NCF2 and between NCF1 and NCF4 was approximately 80% (Fig. 4b and Supplementary Fig. 4c). These data indicate

that NCF1, NCF2, and NCF4 might form identical intracellular puncta after LPS stimulation, which is consistent with the notion that NCF1, NCF2, and NCF4 form a membrane-bound NADPH oxidase after stimulation[38]. Interestingly, sequential LPS and ATP treatment, which triggers NLRP3 inflammasome activation and ASC speck formation, caused a remarkable reduction in puncta colocalization between NCF1 and NCF2 in WT BMDMs compared with the effect of LPS treatment alone (Fig. 4a, b), suggesting that NCF1 and NCF2 are separated in WT BMDMs in response to inflammasome activation. However, the colocalization between NCF1 and NCF2 in *Ncf4*$^{-/-}$, *Asc*$^{-/-}$, and *Nlrp3*$^{-/-}$*Aim2*$^{-/-}$ BMDMs induced by LPS and ATP treatment were comparable or slightly reduced to that induced by LPS alone (Fig. 4a, b), suggesting that the initial inflammasome assembly might have a positive feedback signaling to drive NCF1 separation from NCF2. The colocalization between NCF1 and NCF4 in WT, *Asc*$^{-/-}$ and *Nlrp3*$^{-/-}$*Aim2*$^{-/-}$ BMDMs treated with LPS and ATP was only slightly reduced compared to that of LPS treatment alone (Fig. 4a, b), suggesting that NCF1 and NCF4 mostly maintain their spatial localization with one another.

The colocalization of ASC specks and puncta formed by NCF1, NCF2, and NCF4 was analyzed in WT BMDMs. ASC speck formation induced by LPS and ATP treatment was substantially reduced in *Ncf4*$^{-/-}$, *Asc*$^{-/-}$, and *Nlrp3*$^{-/-}$*Aim2*$^{-/-}$ BMDMs (Supplementary Fig. 4d). Notably, the colocalization between ASC specks and NCF4 puncta was much higher than that between ASC specks and NCF1 and NCF2 (Fig. 4c, d). Collectively, these data indicate that the punctate formation by NCF4, NCF1, and NCF2 triggered by LPS stimulation might contribute to the intracellular localization of ASC specks induced by sequential LPS and ATP stimulation, and NCF4 plays dominant and essential roles in the guidance of ASC speck formation.

### NCF4 balances NADPH oxidase activity and inflammasome activation
NCF4 is a component of the NADPH oxidase complex and functions in the production of ROS within phagocytic cells[33]. We performed a DCFDA analysis and found that LPS plus ATP treatment induced more ROS production than LPS alone in WT BMDMs, and that *Ncf4*$^{-/-}$ BMDMs had a decrease in ROS production in response to LPS plus ATP treatment (Fig. 5a). Furthermore, ROS inhibition using inhibitors consistently reduced NLRP3 inflammasome activation triggered by LPS and ATP treatment (Supplementary Fig. 5a, b). To determine whether the reduced inflammasome activation in *Ncf4*$^{-/-}$ BMDMs was due to impaired ROS production, we treated WT and *Ncf4*$^{-/-}$ BMDMs with H$_2$O$_2$ which rescues the intracellular decrease in ROS. However, a reduction in caspase-1 activation, IL-1β production, and ASC speck formation of *Ncf4*$^{-/-}$ BMDMs remained (Fig. 5b and Supplementary Fig. 5c) and that NOX2 inhibition did not reduce caspase-1 activation (Supplementary Fig. 5d, e). These findings suggest that NCF4, in addition to its role in driving ROS production, plays essential roles downstream of ROS signaling leading to inflammasome activation.

We speculated that membrane-bound NCF4 in the NADPH oxidase complex was shifted to a cytoplasmic orientation during ASC speck formation and that this movement was associated with increased ROS intensity. To further investigate the critical roles of

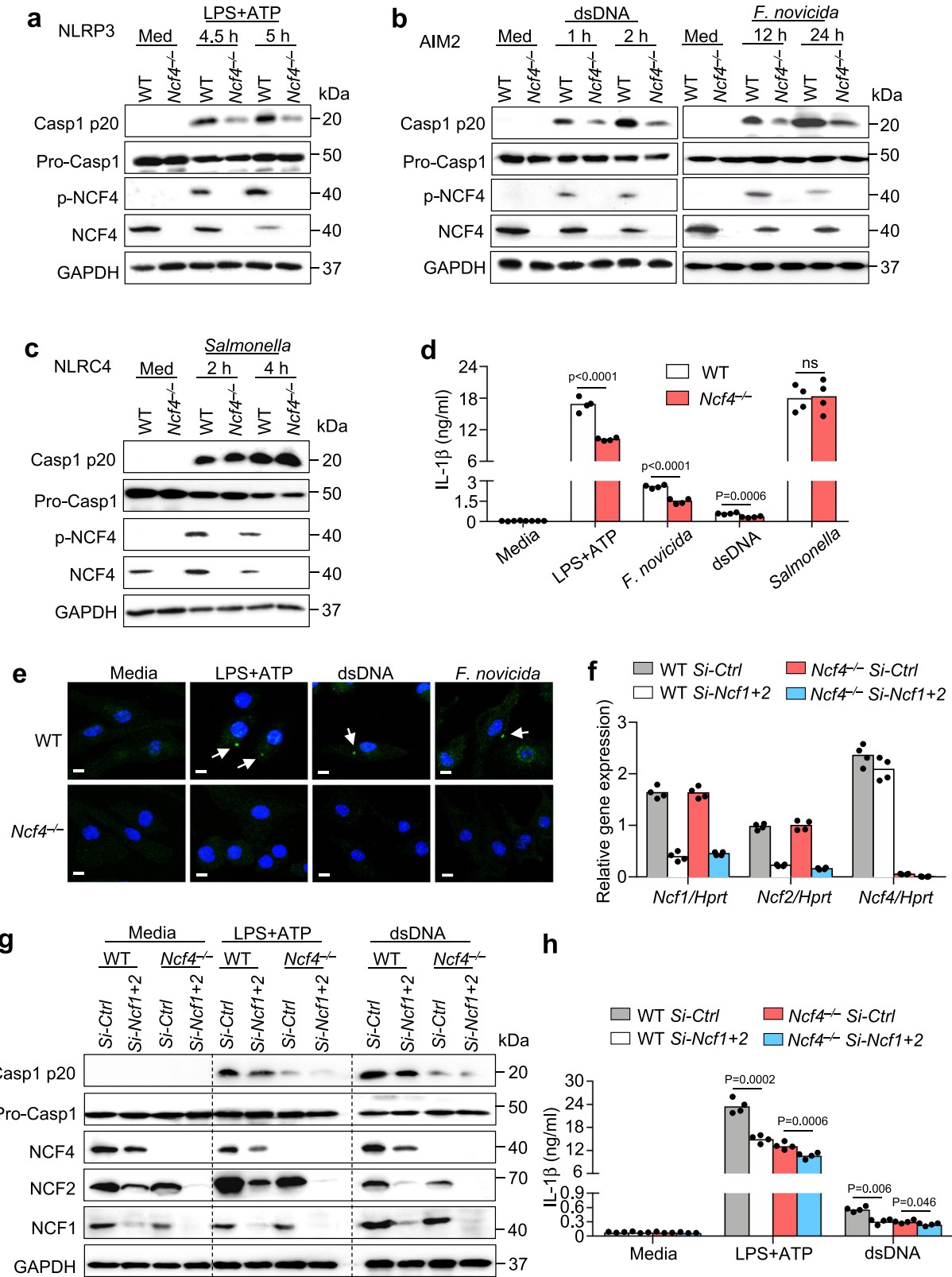

NCF4 in ASC speck formation, we performed a co-IP analysis to identify NCF4- and NCF1-interacting proteins in WT and $Asc^{-/-}$ BMDMs with and without NLRP3 inflammasome activation. We found that, in WT BMDMs treated with LPS and ATP, NCF4 preferentially interacted with NCF1 and ASC, but interacted less with NCF2 or NOX2 (Fig. 5c–f). In $Asc^{-/-}$ BMDMs, the interaction between NCF2 (or NOX2) and NCF4 (or NCF1) was substantially increased compared to that in WT BMDMs

after LPS and ATP treatment (Fig. 5c–f), suggesting that the inflammasome activation signal drives NCF4 (or NCF1) separation from NCF2 (or NOX2). These data indicate that NCF1 and NCF4 separate from NCF2 and NOX2, where NCF1 and NCF4 establish ASC-containing signaling hub driving inflammasome activation.

To confirm the distinct intracellular localization of NCF1, NCF2, and NCF4 after LPS and LPS plus ATP treatments, we performed a 3D

**Fig. 2 | NCF4 mediates the activation of both NLRP3 and AIM2 inflammasomes.**
**a**–**c** Immunoblot analysis of pro-caspase-1 (Pro-Casp1), its subunit p20, total and phosphorylated NCF4 in WT and *Ncf4*$^{-/-}$ BMDMs without treatment (Med) or stimulated with LPS (500 ng/mL, 4 h) and ATP (5 mM, 30 min and 60 min) for NLRP3 inflammasome activation (**a**), transfected with dsDNA (1.5 μg, 1 h and 2 h) or infected with *F. novicida* (200 MOI, 12 h and 24 h) for AIM2 inflammasome activation (**b**), and *Salmonella enterica* Typhimurium (3 MOI, 2 h and 4 h) for NLRC4 inflammasome activation (**c**). **d** Analysis of IL-1β release in WT and *Ncf4*$^{-/-}$ BMDMs without treatment (Media) or stimulated with LPS (500 ng/mL, 4 h) and ATP (5 mM, 60 min), transfected with dsDNA (1.5 μg, 2 h), infected with *F. novicida* (200 MOI, 24 h), and *Salmonella enterica* Typhimurium (3 MOI, 4 h) (*n* = 4 biologically independent samples). **e** Confocal microscopy analysis of ASC speck formation in WT and *Ncf4*$^{-/-}$ BMDMs without treatment or stimulated with LPS (500 ng/mL, 4 h) and ATP (5 mM, 30 min), or transfected with dsDNA (1.5 μg, 1 h); or infected with

*F. novicida* (200 MOI, 12 h) as indicated. Arrows indicate ASC specks. Scale bars: 10 μm. **f**–**h** WT and *Ncf4*$^{-/-}$ BMDMs were transfected with siRNAs of *Ncf1* and *Ncf2*, and further treated with LPS (500 ng/mL, 4 h) and ATP (5 mM, 60 min), and transfected with dsDNA (1.5 μg, 2 h) for inflammasome activation analysis. **f** qRT-PCR analysis of *Ncf1*, *Ncf2*, and *Ncf4* in WT and *Ncf4*$^{-/-}$ BMDMs transfected with control siRNA or siRNAs of *Ncf1* and *Ncf2* (*n* = 4 technical replicates; 3 independent experiments). **g**, **h** Immunoblot analysis of pro-caspase-1 (Pro-Casp1), its subunit p20, NCF4, NCF2, and NCF1 (**g**), and analysis of IL-1β release (**h**) in WT and *Ncf4*$^{-/-}$ BMDMs transfected with siRNAs and further treated with inflammasome stimuli as indicated (*n* = 4 biologically independent samples). Data are from 3 (**d**, **h**) or representative of 3 independent experiments with similar results (**a**–**c**, **e**–**g**). Data represent Mean ± SEM for (**d**, **f**, **h**), 2-sided Student's *t*-test without multiple-comparisons correction, *p*-value is indicated in the graph. Source data are provided as a Source Data file.

constitution analysis of NCF1, NCF2, and NCF4 in WT, *Ncf4*$^{-/-}$, and *Asc*$^{-/-}$ BMDMs. We found that NCF1, NCF2, and NCF4 colocalized to the membrane under LPS-treatment conditions (Fig. 5g and Supplementary Movie 1). In the presence of LPS and ATP treatment, NCF1 and NCF4 preferentially colocalized closer to the nucleus, whereas NCF2 localized largely to the plasma membrane. In comparison, these three proteins colocalized to the membrane in *Asc*$^{-/-}$ BMDMs (Fig. 5g and Supplementary Movie 1). Taken together, these results indicate that NCF4 and NCF1 form puncta to seed the spatial localization of ASC and promote ASC speck formation, while sustaining a source of ROS production to fuel activation of the developing inflammasome complex.

## Phosphorylation of NCF4 is essential for inflammasome activation

Phosphorylation of NCF4 is important for NCF4 translocation to the plasma membrane and activation of NADPH oxidase[39]. To investigate the contribution of NCF4 phosphorylation to inflammasome activation, we treated WT and *Ncf4*$^{-/-}$ BMDMs with the PKC kinase inhibitor midostaurin (PKC412) in conjunction with LPS plus ATP or dsDNA transfection treatments to induce NLRP3 and AIM2 inflammasome activation, respectively. PKC412 inhibited, in a dose-dependent manner, caspase-1 activation and IL-1β secretion in WT BMDMs following NLRP3 and AIM2 inflammasome activation (Fig. 6a, b and Supplementary Fig. 6a, b). In addition, treatment with increasing concentrations of PKC412 correlated with reduced levels of NCF4 phosphorylation and cell death triggered by caspase-1 following NLRP3 inflammasome activation, but not following NLRC4 inflammasome activation (Fig. 6c and Supplementary Fig. 6c). To pinpoint the phosphorylation sites of NCF4 involved in inflammasome activation, we constructed four single phosphorylation site mutations (S85N, T154A, T216M, and S315A) and a quadruple mutation for all four phosphorylation sites (NCF4$^{MT}$) (Supplementary Fig. 6d). We transduced WT NCF4 or its single or quadruple mutants into *Ncf4*$^{-/-}$ BMDMs and assessed NLRP3 inflammasome activation (Fig. 6d). Caspase-1 activation and IL-1β secretion in *Ncf4*$^{-/-}$ BMDMs were rescued in WT NCF4-, NCF4$^{S85N}$-, and NCF4$^{T216M}$-transduced BMDMs, but not in NCF4$^{MT}$-, NCF4$^{T154A}$-, or NCF4$^{S315A}$-transduced BMDMs (Fig. 6d, e). Treatment with LPS alone induced an interaction between WT NCF4 and NOX2, and treatment with LPS plus ATP induced an interaction between WT NCF4 and ASC (Fig. 6f). However, these interactions were inhibited in *Ncf4*$^{-/-}$ BMDMs transduced with NCF4$^{MT}$ (Fig. 6f).

In addition, ASC oligomerization was substantially reduced in *Ncf4*$^{-/-}$ BMDMs compared with WT BMDMs in response to LPS plus ATP (Supplementary Fig. 6e). Notably, *Ncf4*$^{-/-}$ BMDMs transduced with WT NCF4, but not NCF4$^{MT}$, promoted ASC oligomerization in response to NLRP3 activation (Fig. 6g). Furthermore, the puncta formation of NCF4 and NLRP3 or ASC in response to treatment with LPS alone or LPS plus ATP was inhibited in *Ncf4*$^{-/-}$ BMDMs transduced with NCF4$^{MT}$, but not in *Ncf4*$^{-/-}$ BMDMs transduced with WT NCF4 (Fig. 6h and

Supplementary Fig. 6f, g). These results indicate that the phosphorylation of NCF4 is essential for NCF4 puncta seeding and NCF4-mediated inflammasome activation.

## NCF4 is critical for immunosurveillance in the early stages of colorectal cancer progression

The tumor immune microenvironment (TIME) and tumor-infiltrating immune cells are pivotal in regulating tumor growth and cancer metastasis[40]. Although the TIME is a dynamic and heterogeneous network that has been a promising immunotherapeutic target to fight cancer[41], the role of inflammasomes in shaping this environment is not known. To characterize the mechanisms controlled by the NCF4-inflammasome axis in the development of the microenvironment at the precancerous stage, we performed single-cell RNA sequencing analysis of colon tissues from WT and *Ncf4*$^{-/-}$ mice at Day 35 of the 70-day AOM-DSS model (Supplementary Fig. 7a). After eliminating low-quality cells and doublets, a total of 21,196 cells, including 9127 and 12,069 cells, were detected in WT and *Ncf4*$^{-/-}$ colons, respectively. These cells were further subjected to dimensionality reduction and clustering analysis (see "Methods" section). To mitigate the impact resulting from disparate cell counts between the two groups, a downsampling approach was employed to ensure that cell counts in the *Ncf4*$^{-/-}$ colons were equal to those in the WT colons. This step ensured comparability between the two groups for analysis. Overall, we annotated 20 clusters across the two compartments according to the expression of marker genes for each cluster (Fig. 7a, Supplementary Fig. 7b, and Supplementary Data 2). The 20 clusters comprised 13 epithelial cell subsets, including transit-amplifying (TA) cells; multiple enterocyte-associated populations, adenoma-specific cells (ASCs); secretory clusters such as goblet cells, tuft cells, and enteroendocrine cells (EECs); and 7 immune cell subsets (Fig. 7a). Two epithelial cell subsets, enterocyte carcinoma cells (*Mki67*, *Slc12a2*, *Fermt1*) and stem-early enterocyte precursor cells (SeEPCs; *Reg3g*, *Hmgcs2*, *Slc12a2*, *Fermt1*), exhibited stem-like features and were significantly enriched in *Ncf4*$^{-/-}$ colons compared to WT colons (Fig. 7a). Notably, a distinct cluster of epithelial cells characterized by Krt$^+$ (*Krt5*, *Krt6a*, *Krt13*, *Krt14*) was absent in the *Ncf4*$^{-/-}$ colons, and the percentages of two immune cell clusters, CD8$^+$ T and NK cells, were substantially reduced in the *Ncf4*$^{-/-}$ colons (Fig. 7a).

An RNA velocity analysis of all epithelial cells demonstrated a developmental trajectory in WT mice that was initiated by TA cells and involved TA carcinoma and SeEPCs, and this trajectory bifurcated into large intestine gland (Lig) enterocytes and distal enterocyte cells. However, NCF4 deficiency shifted the developmental trajectory from plastic enterocytes to enterocyte carcinoma cells (Fig. 7b). Moreover, high levels of stem-like transcription-related genes (*Slc12a2*, *Fermt1*, *Ncl*, *Nop56* and *Noxa1*) were detected in highly proliferative cell clusters, such as the enterocyte carcinoma, SeEPC and TA clusters (Fig. 7c and Supplementary Fig. 7c). We also saw an increased expression of *Slc12a2*, *Fermt1*, *Ncl*, *Nop56*, and

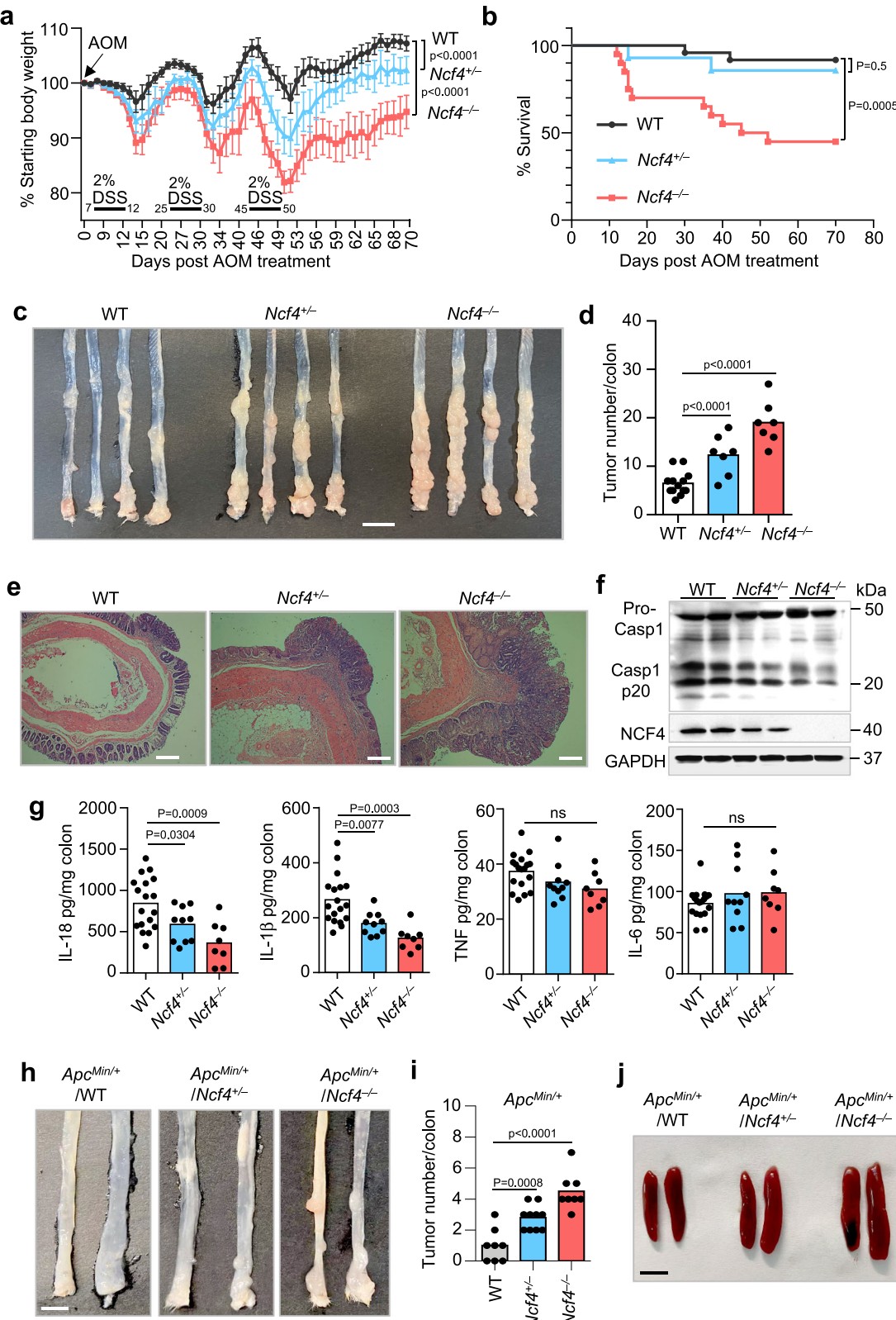

**Fig. 3 | NCF4 deficiency promotes tumorigenesis of CRC. a** Body weight change analysis of gender- and age-matched WT ($n = 14$), $Ncf4^{+/-}$ ($n = 9$), and $Ncf4^{-/-}$ ($n = 13$) mice after AOM injection at Day 0 and three rounds treatment of DSS (2%). **b** Survival analysis of WT ($n = 24$), $Ncf4^{+/-}$ ($n = 14$), and $Ncf4^{-/-}$ ($n = 20$) mice after AOM injection at Day 0 and three rounds of treatment of DSS (2%). **c, d** Colorectal tumors in WT ($n = 13$), $Ncf4^{+/-}$ ($n = 7$), and $Ncf4^{-/-}$ ($n = 7$) mice day 70 under AOM-DSS treatment. Scale bar: 10 mm. **e** H&E staining of Colorectal tumors in WT, $Ncf4^{+/-}$, and $Ncf4^{-/-}$ mice in (**c**). Scale bars: 10 μm. **f, g** Immunoblot analysis of caspase-1 (**f**) and ELISA analysis of IL-18, IL-1β, TNF and IL-6 (**g**) in colon tissues from WT ($n = 18$), $Ncf4^{+/-}$ ($n = 10$), and $Ncf4^{-/-}$

mice ($n = 8$) in (**c**). **h–j** Colorectal tumors (**h**, **i**) and spleens (**j**) from gender- and age-matched offspring of $Apc^{Min/+}$ mice crossed with $Ncf4^{-/-}$ mice with genotype as indicated ($n = 8$ for WT, $n = 10$ for $Ncf4^{+/-}$, and $n = 8$ for $Ncf4^{-/-}$ in **i**). Scale bars: 10 mm for (**h**), and 5 mm for (**j**). Data are from 2 (**b**, **g**, **i**) or representative of 3 independent experiments with similar results (**a**, **c**, **d–f**, **h**, **j**). Data represent Mean ± SEM for (**d**, **g**, **i**), 2-sided Student's *t*-test without multiple-comparisons correction, two-way ANOVA for (**a**), Log-rank (Mantel–Cox) test for (**b**), *p*-value is indicated in the graph. Source data are provided as a Source Data file.

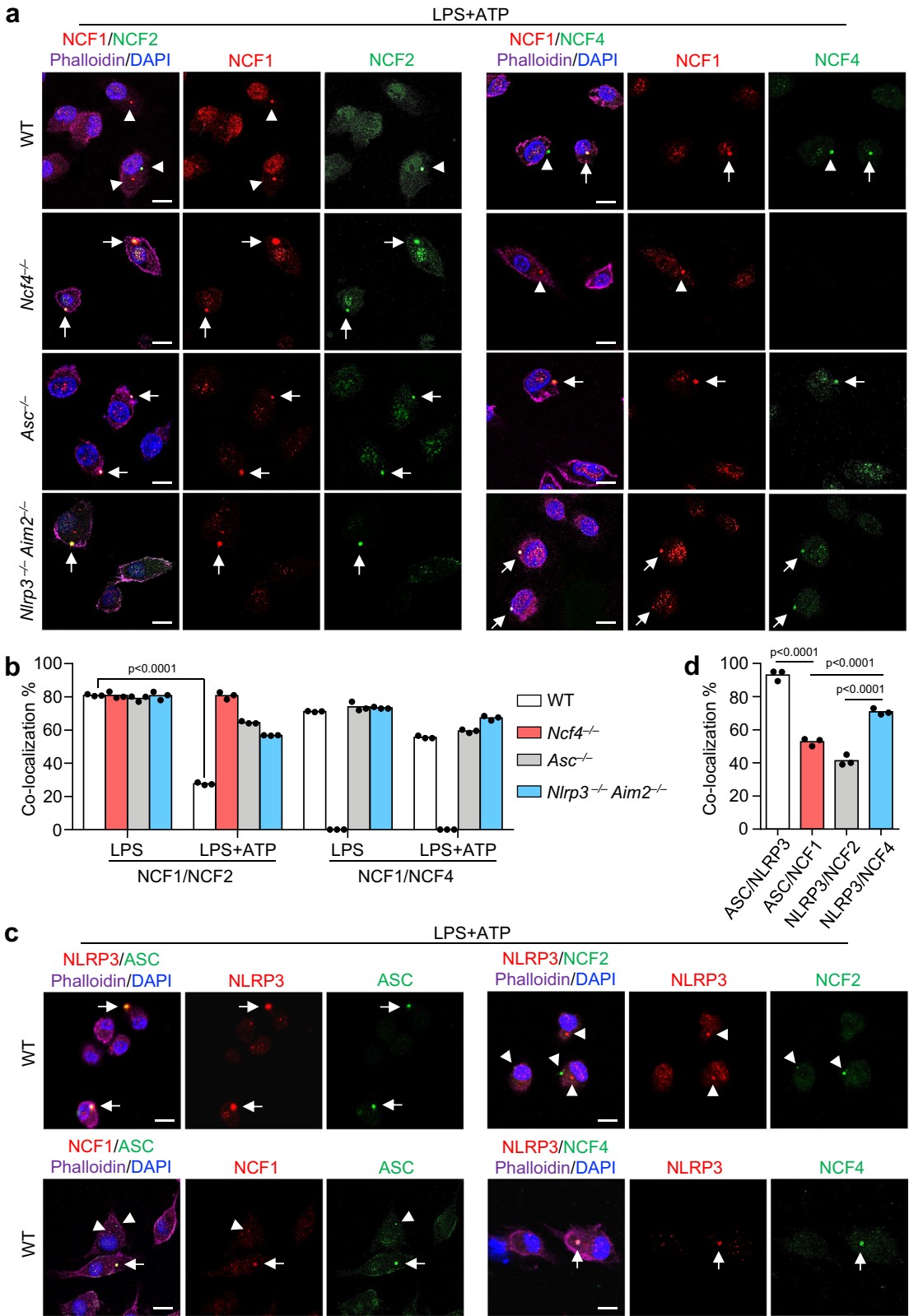

*Noxa1* in the colon tissue of *Ncf4⁻/⁻* mice (Supplementary Fig. 7d). In humans, the expression of *SLC12A2*, *FERMT1*, *NCL*, *NOP56*, and *NOXA1* was significantly upregulated in human tumor cells compared to that in normal controls (Supplementary Fig. 7e). Immunohistochemical staining confirmed an increased expression of SLC12A2 and FERMT1 in the colon tissue of *Ncf4⁻/⁻* mice on Day 35 (Fig. 7d). These results indicate that NCF4 defects cause a

substantial shift in enterocyte-associated clusters from mature enterocytes to plastic enterocytes, enterocyte carcinoma cells, and stem-like precursor cells, which may prevent the development of Krt⁺ epithelial cells important for wound healing and keratinocyte differentiation (Supplementary Fig. 7f).

To chart the immune cell composition and changes in cell states of WT and *Ncf4⁻/⁻* mice, we compared subpopulations of NK and T cells

**Fig. 4 | NCF4 puncta colocalizes with ASC speck. a** Confocal microscopy analysis of the colocalization of NCF1 and NCF2, NCF1, and NCF4 in WT, *Ncf4⁻/⁻*, *Asc⁻/⁻*, and *Nlrp3⁻/⁻Aim2⁻/⁻* BMDMs stimulated with LPS (500 ng/mL, 4 h) and ATP (5 mM, 30 min) for NLRP3 inflammasome activation. Arrows indicate colocalized puncta; Arrowheads indicate separated puncta. Scale bars: 10 μm. **b** Quantification analysis of the colocalization of NCF1 and NCF2, NCF1 and NCF4 in WT, *Ncf4⁻/⁻*, *Asc⁻/⁻*, and *Nlrp3⁻/⁻Aim2⁻/⁻* BMDMs treated with LPS alone, or LPS plus ATP for inflammasome activation. At least 400 cells for NCF1/NCF2 colocalization analysis in LPS plus ATP treated groups and 130 (130–200) cells were analyzed for other groups (*n* = 3 biologically independent samples). **c** Confocal microscopy analysis of the colocalization of NLRP3 and ASC, NCF1 and ASC, NLRP3 and NCF2, NLRP3, and NCF4 in WT BMDMs treated with LPS plus ATP for inflammasome activation. Arrows indicate colocalized puncta; Arrowheads indicate separated puncta. Scale bars: 10 μm. **d** Quantification analysis of the colocalization of NLRP3 and ASC, NCF1 and ASC, NLRP3 and NCF2, NLRP3 and NCF4 in (**c**). At least 130 (130–200) cells were analyzed for each group (*n* = 3 biologically independent samples). Data are from 3 (**b**, **d**) or representative of 3 independent experiments with similar results (**a**, **c**). Data represent Mean ± SEM for (**b**, **d**), 2-sided Student's *t*-test without multiple-comparisons correction, *p*-value is indicated in the graph. Source data are provided as a Source Data file.

that play important roles in the TIME and are activated by inflammasome-associated cytokines. We categorized NK and T cells into eight subclusters, including two clusters of NK cells, one cluster of CD8⁺ T cells, and five clusters of CD4⁺ T cells, according to the expression of specific genes (Fig. 7e, Supplementary Fig. 7g, h and Supplementary Data 3). Notably, the percentages of NK and CD8⁺ T cells were substantially reduced in the *Ncf4⁻/⁻* colon compared to the WT colon (Fig. 7f). A circle plot highlighting the differential number and strength of ligand–receptor (L–R) interactions between WT and *Ncf4⁻/⁻* colons indicated a profound reduction in cell-cell interactions in the absence of NCF4 (Supplementary Fig. 7i). Furthermore, a cell-cell interaction network analysis revealed extensive interactions between CD8⁺ T cells and various cell subsets mediated by IFN-γ signaling in WT but not in *Ncf4⁻/⁻* cells (Fig. 7g). In line with the single-cell RNA sequencing results, a FACS analysis revealed that the proportions of NK and CD8⁺ T cells and IFN-γ-expressing cells were substantially reduced in the colon tissue of *Ncf4⁻/⁻* mice compared to those in WT mice (Fig. 7h, i and Supplementary Fig. 7j). We further verified a reduced production of IL-18, IL-1β and IFN-γ in the absence of NCF4 (Supplementary Fig. 7k). Consistently, the absolute number of IFN-γ-expressing NK cells, CD4⁺ T cells and CD8⁺ T cells was reduced in the colon tissue of *Ncf4⁻/⁻* mice compared to that in WT mice (Supplementary Fig. 8a). IL-18 injection remarkably rescued the reduction in IFN-γ-expressing NK and CD8⁺ T cells in the colon tissue of *Ncf4⁻/⁻* mice treated with AOM-DSS (Supplementary Fig. 8b, c), and attenuated the epithelial dysplasia and tumorigenesis of *Ncf4⁻/⁻* mice treated with AOM-DSS (Supplementary Fig. 8d, e). Collectively, single-cell analyses revealed continual cell transformation from normal to precancerous and cancerous states[42], crucial roles of IFN-γ signaling network in preventing this transformation progress, and NCF4 plays pivotal roles inhibiting this transformation by driving inflammasome-dependent activation of anti-tumor NK and CD8⁺ T cells during early stages of cancer development (Supplementary Fig. 8f).

## Discussion

NCF4 is a component of the NADPH oxidase complex and plays essential roles in ROS production, which mediates bacterial killing and limits neutrophil recruitment in mice with colitis[33,43]. In addition to mediating ROS production, we found NCF4 as a spatiotemporal driver of inflammasome formation within the cell. Mechanistically, NCF4 (p40phox), NCF2 (p67phox), and NCF1 (p47phox) are membrane-bound and colocalize with NOX2 (gp91phox), forming the NADPH oxidase complex for ROS production after LPS stimulation. However, following exposure to an inflammasome activator, NCF4, but not NCF1 or NCF2, predominantly migrates from the membrane to the perinuclear area to induce ASC speck formation and inflammasome activation. Chronic granulomatous disease (CGD) is a primary immunodeficiency characterized by recurrent and invasive infections with bacteria or fungi. Mutations in genes encoding components of the NADPH oxidase complex have been significantly associated with the severity of CGD, Crohn's disease, and cancer[31,32,44–46]. Increasing evidence has demonstrated that CGD patients usually develop intestinal inflammatory diseases, such as Crohn's disease, due to impaired ROS balance and intestinal homeostasis[32,43,47]. A clinical study revealed that patients with

NCF4 deficiency suffered from hyperinflammation and peripheral infections[48]. Thus, our study revealed how the NADPH oxidase subunit NCF4 regulates both NADPH oxidase and inflammasome activation and showed that impaired ROS production and inflammasome activation might play cross-talk in the development of CGD and CD or CRC.

Elevated ROS production is associated with many stress conditions and a common event upstream of the activation of NLRP3 and other inflammasomes. However, the spatiotemporal action of ROS in the regulation of the NLRP3 inflammasome and whether ROS are truly involved in the process of inflammasome activation remain unclear[25]. NADPH oxidase was originally characterized as critical for NLRP3 inflammasome activation when ROS inhibitor treatment was used and in p22phox-knockdown cells[28]. However, caspase-1 activation and IL-1β release were reported to be normal and even increased in macrophages from NADPH oxidase-impaired patients or mice with NADPH oxidase component deficiency, including low levels of membrane-bound p22phox and NOX2 (gp91phox) and cytosolic NCF2 (p67phox)[20,26,29,49–51]. These studies by different groups consistently indicate the dispensable roles of NADPH oxidases in inflammasome activation. Our observation indeed resolves the long-standing problem regarding the profound discrepancies between the roles played by NADPH oxidases and ROS in NLRP3 inflammasome activation.

Excessive ROS production can cause tissue damage induced by oxidative stress. Balanced ROS generation is essential for appropriate immune responses, and limiting ROS production represents an intriguing and potential therapeutic avenue to avoid excess inflammation[27]. Previously, thioredoxin (TRX)-interacting protein (TXNIP) has been reported to interact with NLRP3, leading to inflammasome activation in a ROS-dependent manner[23]. NLRP3 inflammasome was also reported to negatively regulate NADPH oxidase activity through hydrolysis of one or more subunits of the NOX2 complex[52]. Our observation of NCF4 as a ROS sensor that prevents excessive ROS generation and potentiates inflammasome activation after translocating from the membrane-bound NADPH oxidase complex to separate inflammasome-seeding complex reveals a unique ROS regulation mechanism. NCF4 might sense the elevated ROS accumulation derived from both mitochondria and NADPH oxidase, although the precise mechanisms by which ROS and initial inflammasome signaling drive NCF4 relocation between the two signaling complexes need to be ascertained in further studies.

Our single-cell RNA-seq analysis between WT and *Ncf4⁻/⁻* mice revealed critical roles for NCF4 in preventing continuum enterocyte transformation from normal to stem-like and precancerous states by modulating the population and activation of NK and CD8⁺ T cells within the colon tissue microenvironment. We also identified SLC12A2, PERMT1, NOP56, NOXA1, and NCL as potential colon precancerous markers and disease-specific cell subsets and clusters. IL-18 is essential for NK and CD8⁺ T-cell activation and is recognized as a therapeutic target for immunotherapy[36]. Our observations provide alternative approaches for the regulation of inflammasome activation and IL-18 signaling within the colon microenvironment. Heterozygous *Ncf4⁺/⁻* mice also exhibited aggravated colorectal cancer development.

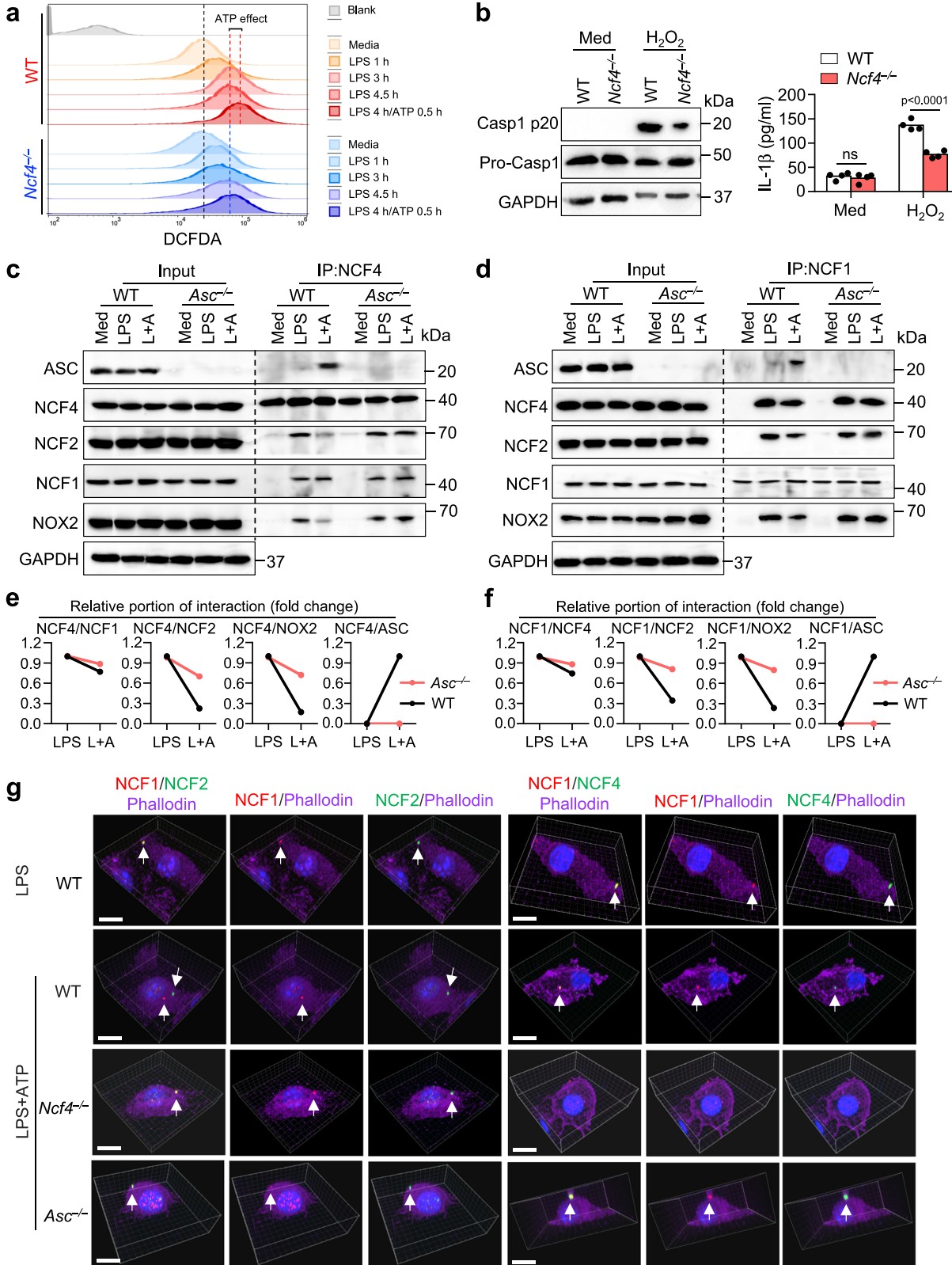

Overall, NCF4 represents a potential target for the therapeutic manipulation of CRC because it orchestrates ROS production and inflammasome activation. Furthermore, we demonstrated the essential role of the NCF4-Inflammasome-IFN-γ axis for the activation of NK and CD8⁺ T cells, which counteract colorectal cancer in the early stages of tumor development.

## Methods

Our research complies with all relevant ethical regulations. The animal experiments were conducted under the approval of the Ethics Committee of Scientific Research of Shandong University (IACUC) with approval numbers ECSBMSSDU2021-2-171 and ECSBMSSDU2020-2-054. The maximal tumor burden permitted by IACUC was 2 cm of

**Fig. 5 | NCF4 contributes to both ROS production and inflammasome activation. a** FACS analysis of ROS production in untreated (Media), LPS-treated (500 ng/mL), and LPS (500 ng/mL) plus ATP (5 mM)-treated WT and $Ncf4^{-/-}$ BMDMs for the indicated time. **b** Immunoblot analysis of caspase-1 maturation and ELISA analysis of IL-1β production ($n = 4$ biologically independent samples) in WT and $Ncf4^{-/-}$ BMDMs without (Med) or with $H_2O_2$ treatment (10 μM, 1 h). **c, d** Co-IP analysis of endogenous NCF4 interacting with ASC and NCF1 (**c**), and endogenous NCF1 interacting with ASC and NCF4 (**d**) in WT and $Asc^{-/-}$ BMDMs without treatment (Med), treated with LPS (500 ng/mL, 4.5 h) or with LPS plus ATP (L+A; LPS, 500 ng/mL, 4 h and ATP, 5 mM, 15 min) for NLRP3 inflammasome activation.

**e, f** Quantification analysis of the relative portion of NCF4 (**e**) and NCF1 (**f**) interaction with other proteins in (**c, d**) ($n = 2$ biologically independent samples). **g** 3D-Confocal microscopy analysis of the colocalization of NCF1 and NCF2, NCF1 and NCF4 in WT BMDMs stimulated with LPS (500 ng/mL, 4.5 h), or in WT, $Ncf4^{-/-}$, and $Asc^{-/-}$ BMDMs stimulated with LPS (500 ng/mL, 4 h) plus ATP (5 mM, 30 min) for NLRP3 inflammasome activation. Scale bars: 10 μm. Data are from 3 (**b**, IL-β) or representative of 3 independent experiments with similar results for others. Data represent Mean ± SEM for (**b**, IL-β), 2-sided Student's t-test without multiple-comparisons correction, p-value is indicated in the graph. Source data are provided as a Source Data file.

tumor diameter, and none of the preclinical trials exceeded this restriction.

## Mice
$Ncf4^{-/-}$ mice were generated by Cyagen Biosciences Inc. Exons 2 through 7 of the $Ncf4$ gene were knocked out by CRISPR-Cas9 system. $Apc^{Min/+}$ mice were purchased from Cyagen Biosciences Inc. $Nlrp3^{-/-}Aim2^{-/-}$ and $Asc^{-/-}$ mice were previously described[53]. WT and knockout mice were SPF-clean and kept under specific pathogen-free conditions in the Animal Resource Center at Shandong University, Jinan, Shandong Province, China. $Ncf4^{-/-}$ mice were backcrossed to C57BL/6 for nine generations. All animal experiments were conducted in accordance with guidelines approved by the Ethics Committee of Scientific Research of Shandong University.

## AOM-DSS model of colorectal tumorigenesis
Male and female mice were used at the age of eight to ten weeks. Gender- and age-matched WT and knockout mice were co-housed for three weeks and separated before injection of azoxymethane or remained co-housed over the course of the experiments. The results were similar in both cases. Mice were injected intraperitoneally with 10 mg/kg of azoxymethane (A5486, Sigma). After 6 days, 1.5-2% DSS (9011181, MP Biomedicals) was given in the drinking water for 6 days followed by regular drinking water for 2 weeks. This cycle was repeated twice more with 1.5-2% DSS and mice were killed on day 70. For day 35 samples, mice were injected with azoxymethane, and after 6 days, fed with 1.5% DSS for 6 days. Mice were then fed with regular water for 14 days and the cycle was repeated once. The recombinant mouse IL-18 was ordered from Sino Biological (50073-MNCE). No randomization or blinding was performed.

## Preparation of BMDMs, treatment, bacterial infection, and siRNA transfection
To generate BMDMs, bone marrow (BM) cells were cultured in L929 cell-conditioned DMEM/F-12 supplemented with 10% FBS, 1% non-essential amino acids, and 1% penicillin-streptomycin for 5 days. The siRNAs target to $Ncf4$, $Ncf1$, $Ncf2$, and $Nox2$ were ordered from RiboBio Company. siRNAs were electroporated into BMDMs by using Neon™ Transfection System (Invitrogen, MPK10025) following the manufacturer's instructions. The siRNA sequences are listed in Supplementary Table 1. Inhibitors NAC (A9165) were ordered from Sigma, PKC412 (120685), DPI (HY-100965), and APO (HY-N0088) were purchased from MedChemexpress. WT, knockout, and siRNA-transfected BMDMs were treated with inhibitors and further stimulated with ligands or infected with bacterial pathogens for indicated times as previously described[6]. The treated and control cells were lysed for RNA and protein analysis.

## Immunoblot analysis and antibodies
Samples were separated by 12% SDS-PAGE, followed by electrophoretic transfer to polyvinylidene fluoride membranes, and membranes were blocked and then incubated with primary antibodies. The following primary antibodies were used: anti-ASC (AdipoGen, AG-25B-0006, 1:1000); anti-V5 (Cell Signaling Technology [CST], 13202, 1:1000); anti-caspase-1 (AdipoGen, AG-20B-0042, 1:3000); anti-NCF4 (Abcam, ab76158, 1:1000); anti-NCF1 (Abcam, ab166930, 1:1000); anti-NCF2 (Abcam, ab109366, 1:1000); anti-p-NCF4 (CST, 4311, 1:1000); anti-p-NCF1 (Affinity, AF3167, 1:1000); anti-p-NCF2 (Affinity, AF4343, 1:1000); anti-NOX2 (Abcam, ab129068, 1:1000); anti-FLAG (Sigma, F3165, 1:5000); anti-Myc (CST, 2278, 1:1000); and anti-GAPDH (CST, 5174, 1:1000). HRP-labeled anti-rabbit, anti-mouse or anti-goat (CST) at 1:10000 dilution was used as the secondary antibody.

## ASC oligomerization
Normal and transduced BMDMs were treated as indicated and then harvested and washed with cold PBS. Cell pellets were lysed in HEPES lysis buffer (30 mM HEPES (pH 7.4), 150 mM NaCl, 1% NP-40, and protease inhibitor cocktails) for 30 min on the ice. Cell lysates were centrifuged at 12,000 rpm for 10 min at 4 °C. The pellet was then cross-linked with DSS (0.5 mM) in HEPES lysis buffer for 30 min at room temperature. The reaction was quenched with 50 mM Tris-HCl (pH 7.4) for 15 min. The cleared, cross-linked samples were boiled in an SDS loading buffer and analyzed by immunoblotting with specific antibodies.

## Immunofluorescence staining and microscopy
For ASC, NCF4, NCF1, NCF2, and NLRP3 immunostaining, treated and untreated BMDMs were fixed in 4% paraformaldehyde for 15 min at room temperature. Cells were washed with PBS and blocked in 1×ELISA buffer with 0.1% saponin for 1 h. Cells were stained with anti-ASC (AdipoGen, AG-25B-0006, 1:200); anti-NLRP3 (AdipoGen, AG-20B-0014, 1:200); anti-NCF4 (Abcam, ab76158, 1:200); anti-NCF1 (Abcam, ab166930, 1:200); anti-NCF2 (Abcam, ab109366, 1:200); or anti-Phallodin (Invitrogen, A22287, 1:500), overnight at 4 °C. Cells were washed, stained with a fluorescence-conjugated secondary antibody (Invitrogen, A-11008, Alexa Fluor™ 488, Goat anti-Rabbit; Invitrogen, A-21422, Alexa Fluor™ 555, Goat anti-Mouse) at 1:300 dilution for 60 min at 37 °C, and mounted using mounting medium (Vector Laboratories, H-1200). Cells were observed on the ZEISS-LSM880 and ANDOR High-speed confocal microscope, ZEISS image acquisition, IMARIS 3-D construction, and data analysis were performed using ZEN black_2-3SP1, ZEN blue 2.6, and IMARIS software.

## IP-MS analysis
WT and $Asc^{-/-}$ BMDMs were infected with $F.$ $novicida$ for 12 h and lysed in IP buffer. ASC antibody was used for immunoprecipitation. The MS experiment and data processing were performed by Novogene Company. Proteins that were specifically identified in WT BMDMs, but not in the $Asc^{-/-}$ BMDMs, are listed in Supplementary Data 1.

## Plasmid transfection and co-IP experiments
The full-length of $Ncf4$, $Ncf1$, $Ncf2$, and $Asc$ was amplified from a mouse cDNA library and subcloned into pCDH and pCMV vectors. Truncated DNA sequences were amplified from full-length of cDNA plasmids and subcloned into a pCDH vector. Site-directed mutations were generated using QuikChange site-directed mutagenesis kits. All plasmids were confirmed by DNA sequencing. The primer

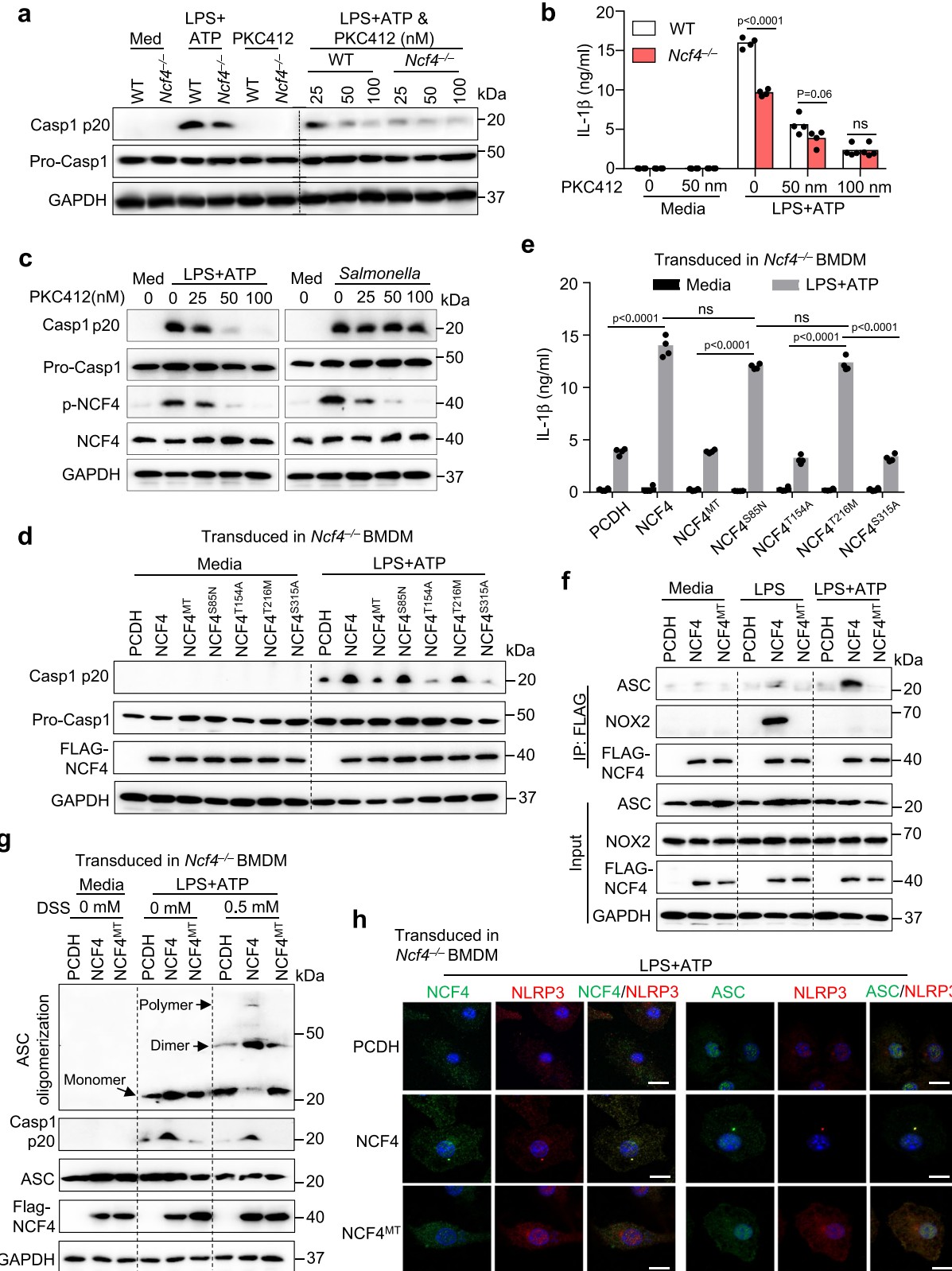

sequences for vector construction are listed in Supplementary Table 1. Lipofectamine 3000 reagents (Invitrogen, Thermo Fisher Scientific) were used for transient transfection of plasmids into HEK293T cells.

For IP, whole HEK293T cells collected 36 h after transfection or BMDMs with and without treatments were lysed in IP buffer composed of 50 mM Tris-HCl (pH 7.4), 2 mM EDTA, 150 mM NaCl, 1% NP-40, 10%

Glycerol, 1 mM DTT, and protease/phosphatase inhibitor cocktails (BioTools). After centrifugation, supernatants were collected and incubated with protein A/G Plus–Agrose (Santa Cruz Biotechnology, sc-2003) and 3 μg of the corresponding antibodies for 12 h at 4 °C, followed by 5 washing with IP buffer. Anti-ASC (AdipoGen, AG-25B-0006, 1:1000), Anti-NCF1 (Abcam, ab166930, 1:1000), and Anti-NCF4 (Abcam, ab76158, 1:1000) were used for endogenous co-IP analysis.

**Fig. 6 | The phosphorylation of NCF4 is important for NLRP3 inflammasome activation. a, b** WT and *Ncf4*⁻/⁻ BMDMs were pretreated with PKC412 (25 nm, 50 nm, and 100 nm) for 1 h, and further stimulated with LPS (500 ng/mL, 4 h) plus ATP (5 mM, 60 min) for NLRP3 inflammasome activation. Immunoblot analysis of pro-caspase-1 (Pro-Casp1) and its subunit p20 (**a**), and analysis of IL-1β release (**b**, *n* = 4 biologically independent samples) in PKC412-treated and untreated BMDMs with and without treatment for NLRP3 inflammasome activation. **c** Immunoblot analysis of phosphorylation of NCF4, pro-caspase-1 (Pro-Casp1) and its subunit p20 in untreated and PKC412-treated WT BMDMs with and without treatment with the NLRP3 activator LPS plus ATP (LPS, 500 ng/mL, 4 h and ATP, 5 mM, 60 min) or the NLRC4 activator *Salmonella enterica* Typhimurium (3 MOI, 2 h). **d, e** Immunoblot analysis of pro-caspase-1 (Pro-Casp1) and its subunit p20 (**d**), and analysis of IL-1β release (**e**, *n* = 4 biologically independent samples) in *Ncf4*⁻/⁻ BMDMs transduced with WT NCF4 and single or quadruple mutations (NCF4ᴹᵀ) of phosphorylation sites of NCF4 as indicated, further stimulated with LPS plus ATP (LPS, 500 ng/mL, 4 h and ATP, 5 mM, 60 min). **f** Co-IP analysis of transduced NCF4

interacting with ASC and NOX2 in *Ncf4*⁻/⁻ BMDMs transduced with control plasmid (PCDH), WT NCF4 and quadruple mutations (NCF4ᴹᵀ) of phosphorylation sites of NCF4 without treatment (Media), treated with LPS (500 ng/mL, 4.5 h) or with LPS plus ATP (LPS, 500 ng/mL, 4 h and ATP, 5 mM, 15 min). **g** ASC oligomerization analysis in *Ncf4*⁻/⁻ BMDMs transduced with control plasmid (PCDH), WT NCF4, and quadruple mutations (NCF4ᴹᵀ) of phosphorylation sites of NCF4 without treatment (Media) and treated with LPS plus ATP (LPS, 500 ng/mL, 4 h and ATP, 5 mM, 30 min). **h** Confocal microscopy analysis NCF4, NCF4ᴹᵀ, NLRP3, and ASC sub-cellular localization in *Ncf4*⁻/⁻ BMDMs transduced with control plasmid (PCDH), WT NCF4 and quadruple mutations (NCF4ᴹᵀ) of phosphorylation sites of NCF4 response to LPS plus ATP (LPS, 500 ng/mL, 4 h and ATP, 5 mM, 30 min). Scale bars: 20 μm. Data are from 3 (**b, e**) or representative of 3 independent experiments with similar results (**a, c, d, f–h**). Data represent Mean ± SEM for (**b, e**), 2-sided Student's *t*-test without multiple-comparisons correction, *p*-value is indicated in the graph. Source data are provided as a Source Data file.

Immunoprecipitated components were eluted by boiling in the SDS loading buffer for 10 min. For immunoblot analysis, immunoprecipitates and input lysates were separated by SDS-PAGE, followed by transfer onto PVDF membranes, and detected by specific antibodies.

## Lentivirus production and infection

The viral particles were prepared by transfecting HEK293T cells with WT and mutated NCF4 plasmids in combination with packaging vectors. Twelve hours later, the media were replaced with fresh complete DMEM. Viral supernatant was harvested and passed through 0.45 μm syringe filter 48 and 72 h after transfection. To establish stably transduced cells, *Ncf4*⁻/⁻ BMDMs were infected 3 times with filtered lentiviral supernatant in the presence of polybrene (8 μg/ml). The transduced cells were cultured in fresh media for further treatments and analysis.

## Single-cell isolation from fresh colon tissues

Colon tissues were isolated from WT and *Ncf4*⁻/⁻ mice and adjacent fat was carefully removed. The colon tissues were longitudinally opened and cut into 1/2-inch pieces, which were then submerged in 20 mL of ice-cold PBS. The tissues were briefly agitated for 30 s, and the dirty PBS was discarded. Each individual tissue was then transferred to a 20 mL solution of pre-warmed (37 °C) EDTA buffer (10% FBS, 25 mM HEPES, 1 mM DTT, 1 mM EDTA, and 100 mg/mL penicillin-streptomycin in PBS), shaken for 15 min (200–220 rpm), and the supernatant was discarded. The tissues were then placed into another conical tube and 30 mL of pre-warmed RPMI-10% FCS (10% FBS, 25 mM HEPES, and 100 mg/mL penicillin-streptomycin in RPMI 1640) was added. The tube was briefly agitated for 30 s, and the lysate was discarded using a cell strainer. Subsequently, the colon tissues were digested in a freshly prepared digestion buffer (1 mg/mL collagenase [Sangon Biotech A004202] and 1 U/mL DNase I [Diamond B002004-0005] in RPMI-10% FCS) at 37 °C on a roto-mixer for 60 min. After digestion, the tissues were gently mixed and the tube was centrifuged at 1500 rpm for 5 min at 4 °C. The supernatant was discarded, and 30 mL of RPMI-10% FCS was added to the tube. The tissues were gently mixed with a 10 mL pipette and filtered through a 70 μm cell strainer into a new conical tube. The tube was centrifuged again at 1500 rpm for 5 min at 4 °C, and the supernatant was discarded. The pellet was carefully resuspended in a final volume of 10 mL of 37.5% Percoll, followed by transfer into a new 15 mL centrifuge tube, and then centrifuged at 1,500 rpm for 10 min at 4 °C with normal acceleration and deceleration. Finally, the supernatant was removed, and the cells were gently resuspended in 1 mL RPMI-10% FCS and transferred to a new 15 mL centrifuge tube for further processing. The single-cell RNA-seq experiments of mouse colon in WT and *Ncf4*⁻/⁻ groups were performed using the Single-Cell 3′ RNA Reagent Kits (10x

Genomics, Pleasanton, California) according to the manufacturer's instruction.

## Single-cell RNA sequencing analysis

The raw scRNA-seq data were firstly mapping to the mouse genome (mm10) and quantified using 10X Genomics Cellranger (v4.0.0) (https://support.10xgenomics.com). Then, the filtered feature-barcode matrix including features, barcode list, and matrix was generated and regarded as input to Seurat (v4.3.0)[54]. To eliminate inferior cells, cells were excluded if they contained a gene count <=200 or >=7000, a mitochondrial gene expression proportion >=0.7, and at least one gene expressed in three cells. The package DoubletFinder (v2.0.3) was used for doublet removal. After performing dimensionality reduction and finding the nearest neighbor (dims = 30), the 'FindClusters' module in Seurat with the resolution 0.7 was employed to generate clusters. The cellular clusters were subsequently manually categorized into distinct cell types based on their marker gene expression. To achieve an equal count of total cells between WT and *Ncf4*⁻/⁻ groups, the *Ncf4*⁻/⁻ group underwent downsampling of cells. In order to identify genes specific to each cell type, the 'FindAllMarkers' function in Seurat was used, applying the following criteria (min.pct = 0.25, logfc.threshold = 0.5, return.thresh = 0.01, test.use=wilcox). Gene functional enrichment analysis was conducted using the clusterProfiler package (v4.6.2)[55]. To analyze the subcluster of immune cells, including NK cells and T cells, the 'FindClusters' function in Seurat was used with a resolution of 0.8, after performing dimensionality reduction and finding the nearest neighbor (dims = 10), to generate cellular clusters. Cell-cell communication was analyzed and inferred using CellChat (v1.6.1) with the default parameters[56]. To calculate the RNA velocity of the single cells, the CellRanger output BAM files were firstly transformed to loom files using velocyto CLI[57], and then calculated and visualized using scVelo[58].

## The Cancer Genome Atlas database (TCGA) data analysis

The expression data and clinical data of Colon and Rectum Adeno-carcinomas (TCGA-COAD, TCGA-READ project) was downloaded from the TCGA website (https://portal.gdc.cancer.gov), Gene read counts were converted to fragments per kilobase per million (FPKM). Expression comparison between tumor and normal tissue, and in tumor-normal matched pairs was performed using the Wilcoxon test after transforming the FPKMs via log2(FPKM+1). The gene expression values log2(FPKM+1) were then displayed using the scatter and box plot. Survival analysis was performed by using R package survival. We used the Cox proportional hazard model to calculate the hazard ratio (HR), and the 95% confidence interval (CI) was reported. The Kaplan–Meier survival curve was modeled by survfit function. We used the maxstat.test function of R package maxstat to identify the best cutting points. This function was used to perform a dichotomy of gene expression and divided the patients into two groups according to the

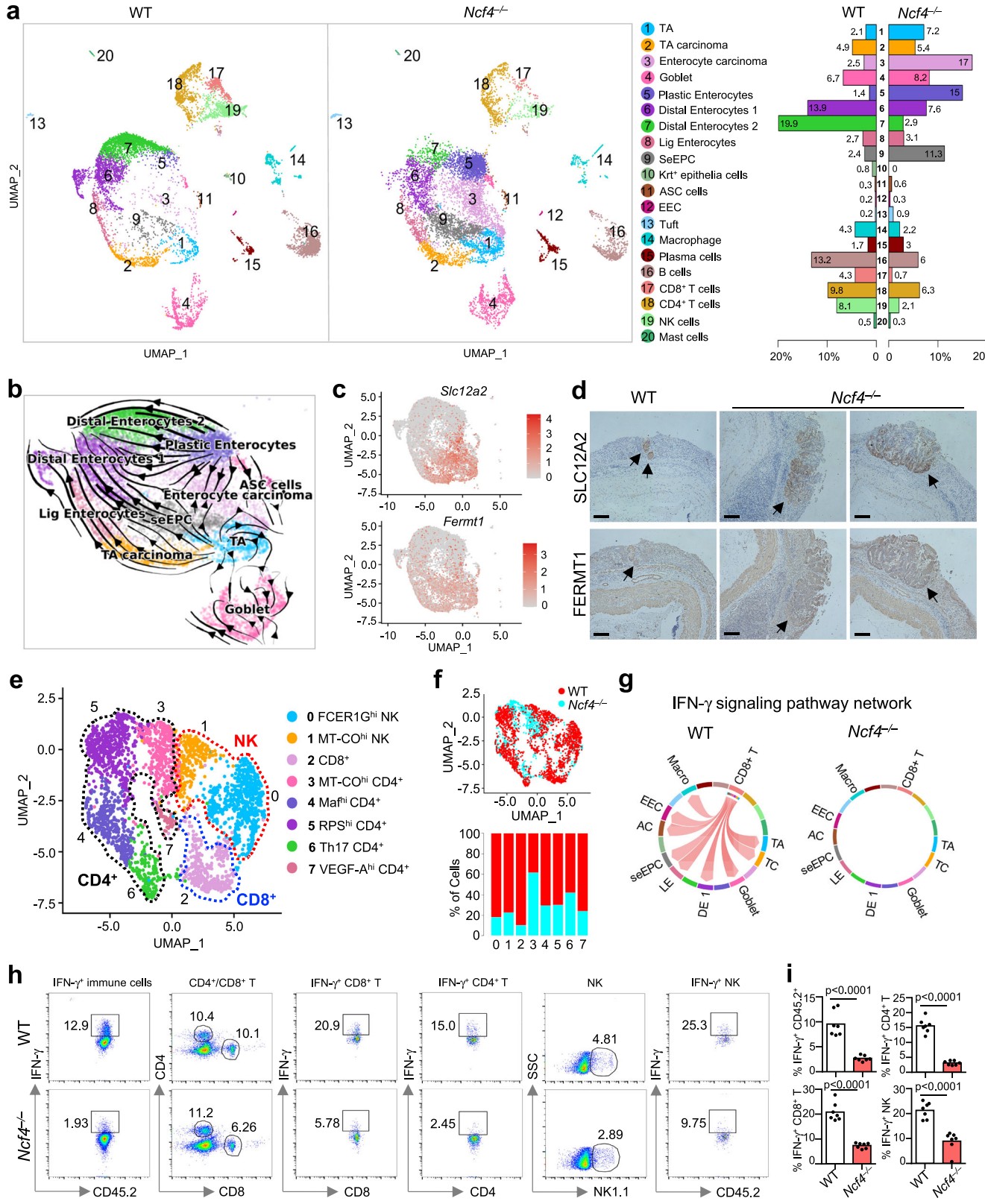

selected maximum logarithm statistics. The two-sided long-rank test was used to compare Kaplan–Meier survival curves.

**Flow cytometry analysis**
For flow cytometric analysis of T cells and NK cells, cells prepared from the colon were stained using a subset of antibodies. Cell preparation and staining with cell surface marker and intracellular cytokine were carried out as described previously[59,60], and cells were analyzed on a BD LSR Fortessa Cell Analyzer (BD Biosciences). For ROS analysis, untreated and treated WT and *Ncf4*−/− BMDMs were stained with DCFDA (Sigma-Aldrich, 287810) for 30 min, and analyzed by FACS.

**Fig. 7 | NCF4 promotes the activation of NK and CD8⁺ T cells within the colonic microenvironment in the precancerous stage.** Single-cell RNA sequencing analysis of colonic cells isolated from WT (n = 3) and *Ncf4⁻/⁻* mice (n = 3) at Day 35 post AOM treatment. **a** Uniform manifold approximation and projection (UMAP) plot and fraction illustrating the distribution of immune and epithelial cells colored by cluster. TA, transit-amplifying cells; Lig Enterocytes, large intestine gland enterocytes; ASC, adenoma-specific cells; EEC, enteroendocrine cells; and SeEPC, stem-early enterocyte precursor cells. **b** UMAP embedding of all epithelial cells with the stochastic representation of the RNA velocity. **c** UMAP plot of epithelial cells showing expression of a gene with stem-like features (tumor marker), *Slc12a2* and *Fermt1*. **d** Representative immunohistochemical image of colon tissues from WT (n = 3) and *Ncf4⁻/⁻* mice (n = 3) stained with anti-SLC12A2 and anti-FERMT1. Scale bars: 10 μm. **e** UMAP embedding of sub-clustering of CD4⁺ T, CD8⁺ T, and NK cell clusters, colored by subclusters. The identity of each cluster was redetermined by

its predominant cells. The annotation of each subcluster was determined by its marker genes. **f** UMAP plot and fraction illustrating the distribution of the subcluster of immune cells in (**e**). The cells in the UMAP plot were colored by sample identity. **g** Chord diagram showing potential interactions or communication between different cell types mediated by IFN-γ signaling pathway. There was no significant interaction in *Ncf4⁻/⁻* mice compared to the WT mice. TA transit-amplifying cells, TC TA carcinoma, DE1 distal enterocytes 1, LE large intestine gland enterocytes, AC adenoma-specific cells, EEC enteroendocrine cells. **h, i** Representative flow cytometry plots (**h**) and quantification analysis (**i**) of IFN-γ⁺ cells in colonic CD45.2⁺, CD4⁺ T, CD8⁺ T and NK cells from WT (n = 7) and *Ncf4⁻/⁻* (n = 7) mice as indicated. Data are representative of 3 independent experiments with similar results (**d, h, i**). Data represent Mean ± SEM for (**i**), 2-sided Student's t-test without multiple-comparisons correction, p-value is indicated in the graph. Source data are provided as a Source Data file.

## Preparation of tissue samples for HE staining
The colon tissues from control and AOM/DSS-treated mice were fixed in 10% formalin, and 5-μm sections were stained with hematoxylin and eosin (H&E) and examined with a microscope.

## Real-time qRT-PCR
Total RNA was isolated from cells and tissues using TRIzol reagent (Invitrogen, Thermo Fisher Scientific). cDNA was reverse transcribed using M-MLV reverse transcriptase (Promega). Real-time qRT-PCR was performed on the Roche LightCycler 96 Real-Time Detection System. The primer sequences are listed in Supplementary Table 1.

## ELISA
The in vivo and in vitro samples were analyzed for cytokine release using ELISA MAX Standard Sets from BioLegend (Mouse IL-1β, 432601; Mouse IL-6, 431301; Mouse TNF, 430901) and Beyotime Biotechnology (Mouse IL-18, PI553) according to the manufacturer's instructions.

## Statistical analyses
Data are presented as the mean ± standard error of the mean (SEM). Statistical analyses were performed using two-way ANOVA, Wilcoxon signed rank, 2-tailed Student's t, and log-rank tests. P-values of 0.05 or less were considered significant.

## Reporting summary
Further information on research design is available in the Nature Portfolio Reporting Summary linked to this article.

## Data availability
The single-cell RNA-seq datasets generated in this study have been deposited at GSA under accession number CRA010882. The proteomics datasets generated in this study are available via ProteomeXchange with identifier PXD052497. The publicly available TCGA RNA-seq data used in this study were downloaded from GDC (TCGA-COAD, TCGA-READ) and is available at (https://portal.gdc.cancer.gov/projects/TCGA-COAD) and (https://portal.gdc.cancer.gov/projects/TCGA-READ). The remaining data are available within the Article, Supplementary Information or Source Data file. Source data are provided with this paper.

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

## Acknowledgements

This work was supported by the National Natural Science Foundation of China (82125021, X.Q., 82301995, L.L., 82321002, X.Q., and 82072255, T.X.), Cutting Edge Development Fund of Advanced Medical Research Institute (GYY2023QY01, X.Q.), and Shandong Province (2022GJJLJRCO2-005, X.Q.).

## Author contributions

X.Q. and L.L. designed the study. L.L., R.M., S.Y., Q.X., J.M., Y.G., S.T., X.X., C.G., H.L., C.M., S.M.M., X.M., T.X., and X.Q. performed experiments and analyzed the data. X.Q., T.X., S.M.M., L.L., and S.Y. wrote the manuscript.

## Competing interests

The authors declare no competing interests.

## Additional information

[1]Key Laboratory for Experimental Teratology of the Ministry of Education, Advanced Medical Research Institute, Qilu Hospital, Cheeloo College of Medicine, Shandong University, Jinan 250012 Shandong, China. [2]CAS Key Laboratory of Genome Sciences and Information, Collaborative Innovation Center of Genetics and Development, Beijing Institute of Genomics, and China National Center for Bioinformation, Chinese Academy of Sciences, Beijing 100101, China. [3]State Key Laboratory of Digital Medical Engineering, School of Biological Science and Medical Engineering, Southeast University, Nanjing, China. [4]Department of Immunology, Key Laboratory for Experimental Teratology of Ministry of Education, Shandong Provincial Key Laboratory of Infection & Immunology, School of Basic Medical Science, Shandong University, Jinan, Shandong, China. [5]Department of Oncology, Affiliated Hospital of Shandong University of Traditional Chinese Medicine, Jinan, Shandong, China. [6]Center for Reproductive Medicine, Cheeloo College of Medicine, Key Laboratory of Reproductive Endocrinology of Ministry of Education, Shandong University, Jinan, China. [7]Division of Immunology and Infectious Disease, The John Curtin School of Medical Research, The Australian National University, Canberra, ACT, Australia. [8]National Key Laboratory for Innovation and Transformation of Luobing Theory; The Key Laboratory of Cardiovascular Remodeling and Function Research, Chinese Ministry of Education, Chinese National Health Commission and Chinese Academy of Medical Sciences, Jinan, Shandong, China. [9]These authors contributed equally: Longjun Li, Rudi Mao, Shenli Yuan. [10]These authors jointly supervised this work: Si Ming Man, Xiangbo Meng, Tao Xu, Xiaopeng Qi. ✉e-mail: siming.man@anu.edu.au; xbmeng@sdu.edu.cn; tao.xu@sdu.edu.cn; xqi@email.sdu.edu.cn

