## [Peer Review File · Nature Communications]

NCF4 attenuates colorectal cancer progression by modulating inflammasome activation and immune surveillanceREVIEWER COMMENTS

Reviewer #1 (Remarks to the Author): with expertise in cancer immunology, inflammasome

In this manuscript, the authors identify an interaction between the NADPH oxidase subunit NCF4 and the inflammasome adaptor protein ASC. They show that following Nlrp3 or Aim2 inflammasome activation, NCF4 co-localizes with ASC spec in macrophages and that loss of Ncf4 impairs ASC spec formation. In vivo, they show that Ncf4^{-/-} mice were susceptible to models of CRC and have decreased inflammasome readouts in the colon.

While the hypothesis is interesting, the paper is descriptive with no mechanistic support of the observed findings.

General comments:

- 1) The authors first unravel an interaction between NCF proteins and ASC by IP-MS and validate this by co-IP experiments in primary BMDMs and by overexpression experiments in HEK293T cells. The co-IP are conducted between NCF proteins and ASC alone. It'd be important to reconstitute the inflammasome components and titrate in NCF proteins to determine a) stoichiometry, b) competition or synergy and c) understand which domain of ASC is involved and how NCF4 and NCF1 organize within the inflammasome complex.
- 2) They show using IF in BMDMs that NCF4 and NCF1 play a role in ASC speck formation. This is very descriptive. No mechanistic details are provided on what directs the cellular re-distribution of these NCF proteins and how they promote inflammasome assembly.
- 3) Figure 6 is problematic. The authors claim that NCF4 phosphorylation is required for inflammasome activation. However, the data with the used kinase inhibitor and with a non-phosphorylatable NCF4 is not convincing. A) The effect of the kinase inhibitor on inhibiting casp1 processing is not at all proportional to its effect on limiting NCF4 phosphorylation (Fig. 6a) e.g., at the 50nM concentration, PKC412 severely inhibited casp1 processing with minimal effect on NCF4 phosphorylation (Fig. 6c), suggesting off-target effects. B) This further seen in Fig. 6b as the inhibitor decreases IL-1b levels in a dose-dep manner in the

Ncf4^{-/-} cells. C) NCF4MT, a non-phosphorylated mutant, that can still interact with ASC (Fig. 6g) fully inhibits casp-1 processing (Fig. 6d). So what is the mechanism? Is the mutant inhibiting ASC oligomerization per se? this can be tested using native gels for instance. Or is it blocking Nlrp3 trafficking? Or is it inhibiting NCF4 migration? Or is it interfering with cytoskeletal events leading to speck formation? Related to this, IL-1b is still abundantly produced in response to LPS+ATP in cells expressing NCF4MT (1ng/ml), a mutant that fully inhibits casp-1 processing (Fig. 6d, e). If NCF4 acts at the level of nucleating the inflammasome complex, one would expect no inflammasome assembly in NCF4MT expressing cells and much blunted IL-1b levels compared to control cells.

5) The authors show that Ncf4^{-/-} mice are more susceptible to AOM-DSS model of colitis-associated CRC and have more CRC tumors when crossed with APCmin mice. The authors show reduced inflammasome activation readouts in vivo, which provides correlative but not causal evidence linking the two pathways. For causality, the authors need to demonstrate that rescuing inflammasome-induced processes, e.g., treatment of the Ncf4^{-/-} mice with low dose rec. IL-18, reverses the susceptibility phenotype.

Reviewer #2 (Remarks to the Author): with expertise in NCFs, ROS, cancer

In this work is shown that Ncf4, accompanied with Ncf1, could enhance the activation of the NLRP3 inflammasome complex through a ROS independent mechanism. There are many important discoveries in the paper but the conclusions are sometimes overstretched and alternative data/interpretations not discussed. In addition, some questions about experimental quality.

1) The intro and discussion is described in a way to fit the authors findings without discussing alternative findings. It is for example well documented that deficiency of each of the NOX2 components, including NCF1 and NCF4, protects again various classical cancer models in mice. It should also be said that the only identified genetic effect (in contrast to the authors claims), is based on identification of the polymorphism a functionally characterized is a SNP with aa replacing function of NCF1. It causes autoimmune disease due to lower ROS production. There are genetic evidence for NCF4, but on the basis of QTL

analysis. The authors only mention the few individuals with monogenic mutations in NCF4 with an atypical CGD development, but the effects could be due to a number of other pathways than the authors discuss.

2) Lack of experimental description; the gene background, genetic purity and use of littermates are not indicated. In one of the experiments it is said that mice in the different groups shared housing. It raises the question about the other performed experiments as this is a surprising statement as a proper experiment should be with shared housing to avoid cage effects. Pathogenic environment is important, it is important to document that the mice, actually are SPF clean with a test protocol, not only saying that the "conditions are SPF (whatever that means). Basically, it seems that standard conditions for mouse experiments have not been described or has not been followed, pls compare with ARRIVE guidelines. It is also for me disturbing that all experiments are shown as representative examples. There is no way to know that these are selected rather than representative. As these are repeated it should be possible to normalize them with the positive control and show all experiments including statistical evaluation, in addition all performed experiment should be shown in suppl data.

3) How do the authors explain the increased DSS colitis? By decreased inflammasome activation? Decreased NK/CD8T infiltration?

4) Could the increased DSS colitis explain the increased tumor susceptibility?

5) It is unclear why decreased inflammasome activation should lead to decreased CD8T and NK infiltration? There is no conclusive evidence that this is the cause of the colon cancer?

6) It is not convincing that the effect is independent of NOX2 induced ROS. Why not use mice deleted of NOX2 and/or Ncf1 and cross it with the Ncf4 ko to exclude a role of ROS and document the dominant effect of NCF4.

7) Could the unique role of NCF4 be dependent on its unique interactions with phospholipids, leading to anchoring in endosomal membranes rather than plasma

membranes. A release of phosphorylated NCF4 could then be adjacent to the formed inflammasomes?

8) The data with RNAseq is to me not significant and not conclusive and lacks follow up experiments.

Reviewer #3 (Remarks to the Author): with expertise in cancer immunology, inflammasome,

The manuscript by Li et al. presents interesting data which reveals a co-operative role for NCF4 in promoting NLRP3 and AIM2 inflammasome activity. Furthermore, this study employs a number of murine colorectal cancer (CRC) models to confirm the involvement of NCF4 in limiting the intestinal adenocarcinoma progression. Findings are therefore of significance to the field.

However, a number of conclusions being made throughout this manuscript are not sufficiently supported by the data presented. For example, the title claims that NCF4 'licences' CRC by orchestrating inflammasome activation and immune surveillance. While the presented data shows that NCF4 promotes NLRP3/AIM2 inflammasome activity, activation still occurs in the absence of NCF4, albeit at to a lesser extent, therefore NCF4 is not essential for inflammasome activity.

Similarly, while some mechanistic insight is elucidated by this study - regarding the potential bifurcation of NCF4/1 and NCF2 localisation, more conclusive evidence is required to confirm these events, and the order of events regarding ROS production, NCF4-ASC interactions and cellular localisation. Much of the evidence regarding NCF4/1 dissociation from NCF2 is based on confocal microscopy - representative images show 1-2 cells per field. The authors state that a minimum of 130 cells were scored for each BMDM stimulation, but the details of scoring system, and an explanation of how the results are presented (three data points/ treatment in co-localisation graphs) was not provided.

Some additional points that I have regarding the results are:

Figure 1 - It is surprising that some of the more convincing data, such as endogenous co-IP

of ASC and NCF4 following LPS+ATP stimulation of BMDM, was not in the main figures? At minimum the interaction between ASC and NCF4 (identified by Mass Spec) should be confirmed by probing the IP lysates (shown in Figure 1a) for NCF4.

- Overexpression studies show that the PX and PB1 domains are important for ASC interaction. What domain of ASC is required for NCF binding?
- Experiments (in Fig 2) reporting role of NCF4 in inflammasome activation following the range of stimuli (i.e. NLRP3, AIM2, NLRC4) would be better in Figure 1 - which should begin characterising the importance of NCF4 for activation. Can the authors examine or hypothesise whether all inflammasomes requiring ASC may also involve NCF4?
- TCGA database analysis is not necessary - the downregulation of NCF4 in CRC, and its lower levels associating with poor survival/ higher tumour grades have already been reported (Ryan B et al. (2013) Int J Cancer).

Figure 4 - please explain how co-localisation analysis and graphs were generated (as referred to earlier). The hypothesis being presented is that NCF4 shifts from membrane bound localisation to cytoplasmic during ASC speck formation. However only LPS+ATP stimulation images are shown in the main Figs. It is important to show some representative LPS stimulations to support hypothesis. Fig 4d - comparing co-localisation of ASC/NCF1 vs NLRP3/NCF2 does not make sense - can these differences be statistically compared? Better to compare ASC/NCF1 vs ASC/NCF2 etc?

- There also appears to be NCF1/NCF2 staining in the nuclear/peri-nuclear regions of the cell also - is there an explanation for this?

The IP expts (Fig 5c,d) - it is difficult to be convinced that NCF1/NCF4 separate from NCF2/NOX2 based on a single timepoint showing a less intense band for NCF2/NOX2 following LPS+ATP stimulation. A timecourse showing a consistent reduction in NCF2 (or Nox2) binding to NCF4 (or NCF1) following the initiation of speck formation would be more convincing here.

Fig 6 - A negative control (eg. NLRC4 activation) should be included in expts using the PKC inhibitor (PKC412), to show that the inhibitory effects of PKC412 are targeting p-NCF4 in this context. Also, full length NCF4 should be included in all blots where phospho-NCF4 is

shown.

The scRNA seq analysis is very interesting, extending the findings from murine CRC models to potentially identify the NCF4-regulated transcriptional factors and immune cell alterations which contribute to the enhanced CRC phenotype of NCF4-deficient mice.

In summary, the data presented here support conclusions that NCF4 contributes to NLRP3 & AIM2 inflammasome activation; and that NCF4 attenuates CRC development. However stating that NCF4 establishes & fuels inflammasome hub formation has not been conclusively demonstrated.

Reviewer #4 (Remarks to the Author): with expertise in colorectal cancer, cancer immunology

In this study, Li et al. provide novel insights into the role NCF4 in the process of inflammasome assembly and activation. Furthermore, the data presented in the manuscript indicate a potentially crucial role of NCF4 in the process of colorectal cancer formation and progression by controlling immune surveillance.

The authors first identified NCF4 as one of the ASC binding proteins. Indeed, the authors have further shown that NCF4 phosphorylation is crucial for inflammasome activation. Moreover, Ncf4-deficient mice showed a higher tumor burden in two independent murine CRC models. Those results are in line with the analysis of TCGA RNA sequencing data in which the authors observed a reduced survival of patients with low NCF4 expression levels. Overall, the manuscript offers novel and interesting insights into the molecular mechanisms of inflammasome activation. However, the link between those mechanistic aspects on a (sub)cellular level and their functional role in immune surveillance and tumor progression is not fully supported by the data. Furthermore, the analysis of the publicly available human data set is not adequately explained in the manuscript. I detail my concerns below:

Major points:

1) The authors use The Cancer Genome Atlas (TCGA) in their analysis in Figure 1. It is unclear how this data set was analyzed - the authors do not mention it in their results section.

2) In Fig. 1f, the authors correlate the expression of NCF4 and NCF1 with tumor stages and claim that “higher tumor stages” are associated with lower expression of NCF. However, the authors selected only the “T” (Tumor size) part of the TNM staging system in their analysis. Choosing a more comprehensive staging system for their analysis (and a clinically more important one) would be more interesting: for instance, the UICC tumor stage (which is based on the TNM staging but not on the “T” alone).

3) Fig. 1g shows that low NCF4 expression is associated with shorter survival in CRC patients. As mentioned above, it is not clear how this data set was analyzed. Did the authors pool the “COAD” (Colon Adenocarcinoma) and READ (rectal adenocarcinoma) data sets of the TCGA to obtain a true “CRC” cohort?

How was the TCGA patient cohort split into “high” and “low” NCF4 expressing patients? Did the authors use a specific cut-off, the median or quartiles?

When performing a fast analysis using the online data analysis platform for the TCGA (Gepia2; <http://gepia2.cancer-pku.cn/#index>; Tang et. al 2017 doi: 10.1093/nar/gkx247), one does not observe a difference in survival when dividing the pooled COAD and READ cohorts by the median NCF4 expression. In general, the authors should explain in detail how they performed their analysis to make it reproducible.

4) Can the results of the TCGA be confirmed in another cohort of patients? This would be important

5) The authors describe a critical role of NCF4 for the immunosurveillance in CRC. However, this conclusion is not fully supported by the data. The authors link two observations (increased colonic tumor development in Ncf4-deficient mice and a decreased infiltration of CD8 and NK cells) and assume a causal link between them. A causal role of CD8 or NK cells for the tumor development in Ncf4-deficient mice has not been proven in any way. Furthermore, it is worth noting that the analysis of infiltrating immune cells was performed using the AOM/DSS model – in which the degree of inflammation was also different between WT and Ncf4-deficient mice and this might well explain differences in the relative amount of distinct immunological populations infiltrating the mucosa. Further mechanistic experiments testing this link are important.

6) Related to the above point and Fig7h-i: have the authors also analyzed the absolute

amount (and not relative percentage) of infiltrating CD8+ T cells and NK cells?

7) The statement that NCF4 regulates the “proliferation” of anti-tumor NK and CD8 T cells (made in the abstract, the results section and in the discussion) is not supported by the data shown in the manuscript. A lower number of NK and CD8 T cells could be related to lower numbers of infiltrating cells (and not their proliferation). Furthermore, and as mentioned above, it is not even clear if the absolute number of those cells is reduced in the tumors of Ncf4-deficient mice.

Minor points

1) The wording in the manuscript is often confusing. One of several examples is the statement “NCF4 is a susceptible gene that is significantly associated with Crohn’s disease and colorectal cancer” (third paragraph in the results section). Did the authors mean “susceptibility gene”? I recommend carefully checking the manuscript for confusing wording.

2) The authors do not provide a reference when referring to the TCGA Data set

- 1) We performed co-IP experiments of NCF4 and ASC or NOX2, ASC oligomerization, and confocal microscopy analysis of NCF4, NLRP3, and ASC speck formation in untreated and activated *Ncf4*^{-/-} BMDMs transduced with either WT or mutated NCF4 carrying specific phosphorylation sites. Our data collectively indicate that phosphorylated NCF4 switched from the NADPH oxidase complex to ASC to promote ASC oligomerization and inflammasome activation in response to inflammasome-activating signals.
- 2) We injected IL-18 into *Ncf4*^{-/-} mice as part of the AOM-DSS model, and found that the NCF4-inflammasome-IL-18-IFN- γ axis is important for NCF4-mediated immune surveillance in the tumor microenvironment.
- 3) We have repeated the dose-dependent experiment and included *Salmonella*-triggered NLRC4 inflammasome activation as a negative control. Treatment of WT BMDMs with 50 nM PKC412 almost completely inhibited the phosphorylation of NCF4 and NLRP3 inflammasome activation, but not NLRC4 inflammasome activation. In addition, the specificity of inflammasome inhibition by PKC412 was also confirmed by our cell death analysis.
- 4) We have clarified our analysis of the TCGA database and removed distracting data from the manuscript.

Please check our responses to all the specific aspects below:

REVIEWER COMMENTS

Reviewer #1 (Remarks to the Author): with expertise in cancer immunology, inflammasome

In this manuscript, the authors identify an interaction between the NADPH oxidase subunit NCF4 and the inflammasome adaptor protein ASC. They show that following Nlrp3 or Aim2 inflammasome activation, NCF4 co-localizes with ASC spec in macrophages and that loss of *Ncf4* impairs ASC spec formation. In vivo, they show that *Ncf4*^{-/-} mice were susceptible to models of CRC and have decreased inflammasome readouts in the colon.

While the hypothesis is interesting, the paper is descriptive with no mechanistic support of the observed findings.

We thank the reviewer who found our study interesting. We have gone to extraordinary efforts to explore the upstream and downstream effects of the interaction between NCF4 and ASC to add more mechanistic insights.

We found that the phosphorylation of NCF4 was important for NCF4 intracellular localization and inflammasome activation, using *Ncf4*^{-/-} BMDMs transduced with either WT NCF4 or NCF4 carrying mutations at specific phosphorylation sites. We also analyzed NLRP3 and ASC speck formation under these conditions using confocal microscopy.

In addition to confocal microscopy analysis and quantification, we also performed co-IP experiments of NCF4 and ASC or NOX2 in WT and *Ncf4*^{-/-} BMDMs, and *Ncf4*^{-/-} BMDMs transduced with either WT NCF4 or NCF4 carrying mutations at specific phosphorylation sites.

To explore the mechanisms of ASC speck formation, we performed ASC oligomerization analysis in WT, *Ncf4*^{-/-} BMDMs, and *Ncf4*^{-/-} BMDMs transduced with either WT NCF4 or NCF4 carrying mutations at specific phosphorylation sites.

Our data collectively indicate that phosphorylated NCF4 relocate from the NADPH oxidase complex to ASC to promote ASC oligomerization and inflammasome activation.

General comments:

1) The authors first unravel an interaction between NCF proteins and ASC by IP-MS and validate this by co-IP experiments in primary BMDMs and by overexpression experiments in HEK293T cells. The co-IP are conducted between NCF proteins and ASC alone. It'd be important to reconstitute the inflammasome components and titrate in NCF proteins to determine a) stoichiometry, b) competition or synergy and c) understand which domain of ASC is involved and how NCF4 and NCF1 organize within the inflammasome complex.

We have performed co-IP experiments between NCF proteins and full-length or truncated ASC proteins. We confirmed the interaction between NCF4 or NCF1 and the ASC PYD domain, and NCF2 with the PYD and CARD domains of ASC. We have added these new data to Fig. 1d and Extended Data Fig. 1b, c.

Fig. 1d

Extended Data Fig. 1b, c

2) They show using IF in BMDMs that NCF4 and NCF1 play a role in ASC speck formation. This is very descriptive. No mechanistic details are provided on what directs the cellular re-distribution of these NCF proteins and how they promote inflammasome assembly.

We show that the phosphorylation of NCF4 is important for NCF4 intracellular localization and inflammasome activation using *Ncf4*^{-/-} BMDMs transduced with either WT NCF4 or NCF4 carrying mutations at specific phosphorylation sites (Fig. 6d,e). We also analyzed NLRP3 and ASC speck formation under these conditions using confocal microscopy (Fig. 6h; Extended Data Fig. 6f).

In addition to confocal microscopy analysis and quantification, we also performed co-IP experiments of NCF4 and ASC or NOX2 in WT and *Ncf4*^{-/-} BMDMs, and *Ncf4*^{-/-} BMDMs transduced either WT NCF4 or NCF4 carrying mutations at specific phosphorylation sites (Fig. 5c-f; Fig. 6f).

To explore the mechanisms of ASC speck formation, we performed ASC oligomerization analysis in WT, *Ncf4*^{-/-} BMDMs, and *Ncf4*^{-/-} BMDMs transduced with either WT NCF4 or NCF4 carrying mutations at specific phosphorylation sites (Fig. 6g; Extended Data Fig. 6e).

Our data collectively indicate that phosphorylated NCF4 relocate from the NADPH oxidase complex to ASC to promote ASC oligomerization and inflammasome activation. We have added these data to Fig. 6d, e, f, g, h; Extended Data Fig. 6e, f.

Fig. 6d, e: Phosphorylation of NCF4 promotes inflammasome activation.

Fig. 6d

Fig. 6e

Fig. 6h and Extended Data Fig. 6f: Phosphorylation of NCF4 promotes ASC speck formation.

Fig. 6h

Extended Data Fig. 6f

Fig. 6f: Phosphorylation of NCF4 promotes the interaction between ASC and NCF4.

Fig. 6g and Extended Data Fig. 6e: Phosphorylation of NCF4 promotes ASC oligomerization.

Fig. 6g

Extended Data Fig. 6e

3) Figure 6 is problematic. The authors claim that NCF4 phosphorylation is required for inflammasome activation. However, the data with the used kinase inhibitor and with a non-phosphorylatable NCF4 is not convincing. A) The effect of the kinase inhibitor on inhibiting casp1 processing is not at all proportional to its effect on limiting NCF4 phosphorylation (Fig. 6a) e.g., at the 50nM concentration, PKC412 severely inhibited casp1 processing with minimal effect on NCF4 phosphorylation (Fig. 6c), suggesting off-target effects.

B) This further seen in Fig. 6b as the inhibitor decreases IL-1b levels in a dose-dep manner in the *Ncf4*^{-/-} cells.

We have repeated the dose-dependent experiment and included *Salmonella*-triggered NLR4 inflammasome activation as a negative control. Treatment of WT BMDMs with 50 nM PKC412 almost completely inhibited the phosphorylation of NCF4 and NLRP3 inflammasome activation, but not NLR4 inflammasome activation. In addition, the specificity of inflammasome inhibition by PKC412 was also confirmed by cell death analysis. We have added these new data to Fig. 6c and Extended Data Fig. 6c.

Fig. 6c

Extended Data Fig. 6c

C) NCF4MT, a non-phosphorylated mutant, that can still interact with ASC (Fig. 6g) fully inhibits casp-1 processing (Fig. 6d). So what is the mechanism?

Our data show that endogenous interaction between NCF4MT and ASC was substantially reduced compared with that between WT NCF4 and ASC (Figure 6g in previous submission). We repeated this experiment and confirmed the reduced interaction between NCF4MT and ASC. We have added the data to Fig. 6f.

Fig. 6f

Is the mutant inhibiting ASC oligomerization per se? this can be tested using native gels for instance. Or is it blocking Nlrp3 trafficking? Or is it inhibiting NCF4 migration? Or is it interfering with cytoskeletal events leading to speck formation? Related to this, IL-1b is still abundantly produced in response to LPS+ATP in cells expressing NCF4MT (1ng/ml), a

mutant that fully inhibits casp-1 processing (Fig. 6d, e). If NCF4 acts at the level of nucleating the inflammasome complex, one would expect no inflammasome assembly in NCF4MT expressing cells and much blunted IL-1b levels compared to control cells.

We thank the reviewer for the suggestions.

1) We performed ASC oligomerization analysis in WT and *Ncf4*^{-/-} BMDMs, and *Ncf4*^{-/-} BMDMs transduced with either WT NCF4 or NCF4 carrying mutations at specific phosphorylation sites. We found that the absence of NCF4 or non-phosphorylatable NCF4 mutant led to a reduction in the amount of ASC oligomerization. We have added these new data to Fig. 6g and Extended Data Fig. 6e.

Extended Data Fig. 6e

Fig. 6g

2) We performed co-IP analysis of endogenous NCF4 and NOX2 or ASC in *Ncf4*^{-/-} BMDMs transduced with WT or non-phosphorylatable NCF4. We found that NCF4 interacted with NOX2 following LPS stimulation alone, whereas NCF4 interacted with ASC following treatment with LPS plus ATP. This important data indicates that NCF4 migrates from NOX2 to ASC in response to inflammasome activation. However, the non-phosphorylatable NCF4 did not interact with NOX2 or ASC in LPS or LPS plus ATP treatment, suggesting that the phosphorylation of NCF4 is essential for NCF4 trafficking and subcellular localization. We have added these data to Fig. 6f.

Fig. 6f

3) We further performed confocal microscopy analysis to localize NCF4, ASC, and NLRP3 in *Ncf4*^{-/-} BMDMs transduced with WT NCF4 or non-phosphorylatable NCF4. NCF4 but not non-phosphorylatable NCF4 (NCF4^{MT}) colocalized with NLRP3/ASC during inflammasome activation. We have added the data to Fig. 6h and Extended Data Fig. 6f.

Extended Data Fig. 6f

Fig. 6h

4) Regarding how the binding between NCF4 and ASC regulates ASC oligomerization, we feel that more advanced techniques and molecular investigations such as single-molecule tracking and structural biology in future studies would be required to delineate the downstream effects resulting from the interaction between NCF4 and ASC.

5) The efficiency of lentiviral transfection in primary BMDMs is much lower compared with that in other cell lines. We have gone to extraordinary efforts to achieve optimal transfected cells to perform further experiments. We have now repeated the experiment in transduced *Ncf4*^{-/-} BMDMs complemented with WT NCF4 or NCF4 carrying mutations in one or more phosphorylation sites. In line with our previous results, we consistently found that BMDMs carrying NCF4^{MT}, NCF4T154A, and NCF4S315A had reduced caspase-1 activation and IL-1 β production. We have added these new data to Fig. 6d, e.

Fig. 6d

Fig. 6e

4) The authors show that *Ncf4*^{-/-} mice are more susceptible to AOM-DSS model of colitis-associated CRC and have more CRC tumors when crossed with APC^{min} mice. The authors show reduced inflammasome activation readouts in vivo, which provides correlative but not causal evidence linking the two pathways. For causality, the authors need to demonstrate that rescuing inflammasome-induced processes, e.g., treatment of the *Ncf4*^{-/-} mice with low dose rec. IL-18, reverses the susceptibility phenotype.

We have followed the reviewer's advice and injected recombinant IL-18 into *Ncf4*^{-/-} mice treated with AOM-DSS (Extended Data Fig. 7a). Indeed, we found that IL-18 injection rescued the reduced activation of CD8⁺ T cells and NK cells in AOM-DSS-treated *Ncf4*^{-/-} mice, indicating that impaired inflammasome activation in *Ncf4*^{-/-} mice leads to a dysregulated tumor microenvironment. We have added these new data to Extended Data Fig. 8b, c.

Extended Data Fig. 8b, c

Reviewer #2 (Remarks to the Author): with expertise in NCFs, ROS, cancer

In this work is shown that Ncf4, accompanied with Ncf1, could enhance the activation of the NLRP3 inflammasome complex through a ROS independent mechanism. There are many important discoveries in the paper but the conclusions are sometimes overstretched and alternative data/interpretations not discussed. In addition, some questions about experimental quality.

We thank the reviewer found our study important. We have revised our manuscript and added alternative explanations in our manuscript.

1) The intro and discussion is described in a way to fit the authors findings without discussing alternative findings. It is for example well documented that deficiency of each of the NOX2 components, including NCF1 and NCF4, protects against various classical cancer models in mice. It should also be said that the only identified genetic effect (in contrast to the authors claims), is based on identification of the polymorphism a functionally characterized is a SNP with aa replacing function of NCF1. It causes autoimmune disease due to lower ROS production. There are genetic evidence for NCF4, but on the basis of QTL analysis. The authors only mention the few individuals with monogenic mutations in NCF4 with atypical CGD developments, but the effects could be due to a number of other pathways than the authors discuss.

We thank the reviewer for these suggestions. We have now toned down the claim of the role of NCF4 in human diseases in both the introduction and discussion sections. We have added other reports and changed the description to emphasize that the dysregulation of ROS production and inflammasome activation might both involved in the development of CGD, CD, and CRC.

2) Lack of experimental description; the gene background, genetic purity and use of littermates are not indicated. In one of the experiments is said that mice in the different groups shared housing. It raises the question about the other performed experiments as this is a surprising statement as a proper experiment should be with shared housing to avoid cage effects. Pathogenic environment is important, it is important to document that the mice, actually are SPF clean with a test protocol, not only saying that the "conditions are SPF (whatever that means). Basically, it seems that standard conditions for mouse experiments have not been described or has not been followed, pls compare with ARRIVE guidelines. Its also for me disturbing that all experiemnts are shown as representative examples. There is no way to know that these are selected rather than representative. As these are repeated it should be possible to normalize them with the positive control and show all experiments including statistical evaluation, in addition all performed experiment should be shown in suppl data.

We thank the reviewer for these suggestions. We have added more information describing the gene background, genetic purity, use of littermates, experimental description, and source data. We used gender- and age-matched SPF mice for our in vivo experiments.

3) How to the authors explain the increased DSS colitis? By decreased inflammasome activation? Decreased NK/CD8T infiltration?

The activation of NLRP3 inflammasome and downstream IL-18 production has been shown to play protective roles in DSS-induced colitis and colitis-associated colorectal cancer (1-4). The inflammasome-caspase-1-IL-18 axis was considered to mediate barrier integrity and tissue repair in the intestine, which contributes to intestinal homeostasis and the protective role of inflammasome in DSS-induced colitis(5).

4) Could the increased DSS colitis explain the increased tumor susceptibility?

Chronic inflammation is a risk factor for tumorigenesis, yet the precise mechanism of this association is currently unknown. The current thinking within the field is that chronic inflammation impairs wound healing and promotes DNA damage and cell proliferation, all of which ultimately increase the risk of tumor development. The increased DSS-induced colitis in *Ncf4*^{-/-} mice is consistent with the increased tumor susceptibility of *Ncf4*^{-/-} mice in the AOM-DSS model.

5) Its unclear why decreased inflammasome activation should lead to decreased CD8T and NK infiltration? There are no conclusive evidence that this is the cause of the colon d cancer?

IL-18 is a member of the IL-1 family that is secreted followed inflammasome activation, and has an ability to induce IFN- γ production (6). IL-18 can stimulate NK cells and T cells for augmentation of antitumor immunity (7, 8). Therefore, impaired activation of the inflammasome and IL-18 secretion owing to NCF4 deficiency may contribute to the decreased recruitment of activated CD8⁺ T cells and NK cells, which promotes more elevated tumorigenesis.

Importantly, we found that IL-18 injection can indeed restore the activation of CD8⁺ T cells and NK cells. These data indicate that IL-18 bridges inflammasome activation and infiltration of activated CD8⁺ T cells and NK cells. We have added these new data to Extended Data Fig. 8b, c.

Extended Data Fig. 8b, c

6) Its not convincing that the effect is independent of NOX2 induced ROS. Why not use mice deleted of NOX2 and/or Ncf1 and cross it with the Ncf4 ko to exclude a role of ROS and document the dominant effect of NCF4.

We thank the reviewer for this suggestion. We agree that the claim of ROS-independent function should be toned down, and have therefore, removed the mention of “ROS-independent function”.

7) Could the unique role of NCF4 be dependent on its unique interactions with phospholipids, leading to anchoring in endosomal membranes rather than plasma membranes. A release of phosphorylated NCF4 could then be adjacent to the formed inflammasomes?

We thank the reviewer for this suggestion. A previous study reported that the PX domain of NCF1 binds phosphatidylinositol 3,4-bisphosphate and phosphatidic acid, which are mainly found in plasma membrane, whereas the PX domain of NCF4 specifically binds to Ptdins3P mainly found in the endosomal membrane (9). We agree with the reviewer that this important observation may suggest that NCF4 has a more dominant role in promoting inflammasome activation compared with NCF1. However, further mechanisms need to be examined in future studies.

8) The data with RNAseq is to me not significant and not conclusive and lacks follow up experiments.

We apologize for not convincingly explaining the significance of the RNAseq data. Our RNAseq data was key to identifying that at the earlier stages of AOM-DSS model, NCF4 deficiency resulted in decreased activation of CD8⁺ T cells and NK cells, which are two major effector cells that kill tumor cells. We further validated this observation using flow cytometry (Figure 7h, i).

Reviewer #3 (Remarks to the Author): with expertise in cancer immunology, inflammasome,

The manuscript by Li et al. presents interesting data which reveals a co-operative role for NCF4 in promoting NLRP3 and AIM2 inflammasome activity. Furthermore, this study employs a number of murine colorectal cancer (CRC) models to confirm the involvement of NCF4 in limiting the intestinal adenocarcinoma progression. Findings are therefore of significance to the field.

We thank the reviewer found our study interesting and significant.

However, a number of conclusions being made throughout this manuscript are not sufficiently supported by the data presented. For example, the title claims that NCF4 'licences' CRC by orchestrating inflammasome activation and immune surveillance. While the presented data shows that NCF4 promotes NLRP3/AIM2 inflammasome activity, activation still occurs in the absence of NCF4, albeit at to a lesser extent, therefore NCF4 is not essential for inflammasome activity.

We have changed the title to “NCF4 inhibits colorectal cancer by modulating inflammasome activation and immune surveillance”.

Similarly, while some mechanistic insight is elucidated by this study - regarding the potential bifurcation of NCF4/1 and NCF2 localisation, more conclusive evidence is required to confirm these events, and the order of events regarding ROS production, NCF4-ASC interactions and cellular localisation. Much of the evidence regarding NCF4/1 dissociation from NCF2 is based on confocal microscopy - representative images show 1-2 cells per field. The authors state that a minimum of 130 cells were scored for each BMDM stimulation, but the details of scoring system, and an explanation of how the results are presented (three data points/ treatment in co-localisation graphs) was not provided.

We have added more detailed information describing our quantification analysis of confocal microscopy images. We analyzed data from three independent repeats (revised figure legends and added source data).

In addition to confocal microscopy analysis, we also performed co-IP experiments of endogenous NCF4 and ASC or NOX2 in WT and *Ncf4*^{-/-} BMDMs, and *Ncf4*^{-/-} BMDMs transduced with a WT NCF4 or a non-phosphorylatable NCF4 (Fig. 5c-f; Fig. 6f). We also found that NCF4 interacted with NOX2 in response to LPS stimulation, whereas NCF4 interacted with ASC in response to treatment with LPS plus ATP, indicating that NCF4 migrates from NOX2 to ASC in response to inflammasome activation. However, the non-phosphorylatable NCF4 did not interact with NOX2 or ASC in response to LPS alone or LPS plus ATP, suggesting that phosphorylation of NCF4 is essential for NCF4 trafficking and subcellular localization. We have added these new data to Fig. 5c-f, and Fig.6f.

Fig. 5c-f

Fig.6f

Some additional points that I have regarding the results are:

Figure 1 - It is surprising that some of the more convincing data, such as endogenous co-IP of ASC and NCF4 following LPS+ATP stimulation of BMDM, was not in the main figures? At minimum the interaction between ASC and NCF4 (identified by Mass Spec) should be confirmed by probing the IP lysates (shown in Figure 1a) for NCF4.

We thank the reviewer for this suggestion. We have moved the endogenous co-IP data to Fig. 1e.

Fig. 1e

- Overexpression studies show that the PX and PB1 domains are important for ASC interaction. What domain of ASC is required for NCF binding?

We performed new co-IP experiments using full-length NCF1, NCF2, and NCF4, and full-length or truncated ASC to identify the interaction domain of ASC. We show that NCF4 or NCF1 interacted with the ASC PYD domain; and NCF2 interacted with both the PYD and CARD domains of ASC. We have added these new data to Fig. 1d and Extended Data Fig. 1b, c.

Fig. 1d

Extended Data Fig. 1b, c

- Experiments (in Fig 2) reporting role of NCF4 in inflammasome activation following the range of stimuli (i.e. NLRP3, AIM2, NLRC4) would be better in Figure 1 - which should begin characterising the importance of NCF4 for activation. Can the authors examine or hypothesise whether all inflammasomes requiring ASC may also involve NCF4?

We performed new ASC oligomerization analysis and found that NCF4 phosphorylation was important for ASC oligomerization (Fig. 6g and Extended Data Fig. 6e). Given that NLRP3, AIM2 and NLRC4 are the best characterized inflammasomes, and that other inflammasomes are not fully characterized in terms of their modes of activation or expressed in BMDMs, we feel that investigations into NCF4 in these emerging inflammasomes, such as NLRP6 and NLRP10, would be better suited for future studies.

Extended Data Fig. 6e

Fig. 6g

- TCGA database analysis is not necessary - the downregulation of NCF4 in CRC, and its lower levels associating with poor survival/ higher tumour grades have already been reported (Ryan B et al. (2013) Int J Cancer).

We followed the reviewer's advice and removed the data.

Figure 4 - please explain how co-localisation analysis and graphs were generated (as referred to earlier). The hypothesis being presented is that NCF4 shifts from membrane bound localisation to cytoplasmic during ASC speck formation. However only LPS+ATP stimulation images are shown in the main Figs. It is important to show some representative LPS stimulations to support hypothesis. Fig 4d - comparing co-localisation of ASC/NCF1 vs NLRP3/NCF2 does not make sense - can these differences be statistically compared? Better to compare ASC/NCF1 vs ASC/NCF2 etc?

We have added more detailed information describing our quantification analysis of confocal microscopy images. We analyzed data from three independent repeats (revised figure legends and added source data). The reason that we compared ASC/NCF1 vs NLRP3/NCF2, but not ASC/NCF1 vs ASC/NCF2 is due to issues of cross-reactivity between species in our available antibodies. Instead, we performed the confocal microscopy analysis of NCF4, ASC, and NLRP3 in *Ncf4*^{-/-} BMDMs transduced with a WT NCF4 or a non-phosphorylatable NCF4. We found NCF4, but not a non-phosphorylatable NCF4 (NCF4^{MT}), colocalized with NLRP3/ASC during inflammasome activation. Stimulation with LPS alone did not induce ASC/NLRP3 speck formation. We have added these new data to Fig. 6h and Extended Data Fig. 6f.

Extended Data Fig. 6f

Fig. 6h

- There also appears to be NCF1/NCF2 staining in the nuclear/peri-nuclear regions of the cell also - is there an explanation for this?

Nuclear or peri-nuclear NCF1 and NCF2 may have additional uncharacterized functions not described in this current manuscript.

The IP expts (Fig 5c,d) - it is difficult to be convinced that NCF1/NCF4 separate from NCF2/NOX2 based on a single timepoint showing a less intense band for NCF2/NOX2 following LPS+ATP stimulation. A timecourse showing a consistent reduction in NCF2 (or Nox2) binding to NCF4 (or NCF1) following the initiation of speck formation would be more convincing here.

We experience technical challenges in endogenous IP experiments because LPS-primed BMDMs are treated with ATP for 15 min before sample collection to avoid

excessive pyroptosis and therefore a loss of cytosolic content. Therefore, it is technically not feasible to add additional time points for a 15 min assay. Instead, we performed co-IP analysis in *Ncf4*^{-/-} BMDMs transduced with a WT NCF4 or a non-phosphorylatable NCF4. We found that NCF4 interacted with NOX2 in response to LPS, and NCF4 interacted with ASC in response to LPS plus ATP. These new data suggest that NCF4 potentially separates from NOX2 and moves to ASC following inflammasome activation by LPS plus ATP. We have added these new data to Fig. 6f. Fig. 6f

Fig 6 - A negative control (eg. NLRC4 activation) should be included in expts using the PKC inhibitor (PKC412), to show that the inhibitory effects of PKC412 are targeting p-NCF4 in this context. Also, full length NCF4 should be included in all blots where phospho-NCF4 is shown.

We thank the reviewer for this great suggestion.

We have repeated the dose-dependent experiment and included *Salmonella*-triggered NLRC4 inflammasome activation as a negative control. Treatment of WT BMDMs with 50 nM PKC412 almost completely inhibited the phosphorylation of NCF4 and NLRP3 inflammasome activation, but not NLRC4 inflammasome activation. We also included full-length NCF4 in the blot. We have added these new data to Fig. 6c.

Fig. 6c

The scRNA seq analysis is very interesting, extending the findings from murine CRC models to potentially identify the NCF4-regulated transcriptional factors and immune cell alterations which contribute to the enhanced CRC phenotype of NCF4-deficient mice.

We thank the reviewer for this comment.

In summary, the data presented here support conclusions that NCF4 contributes to NLRP3 & AIM2 inflammasome activation; and that NCF4 attenuates CRC development. However, stating that NCF4 establishes & fuels inflammasome hub formation has not been conclusively demonstrated.

We have removed this statement.

Reviewer #4 (Remarks to the Author): with expertise in colorectal cancer, cancer immunology

In this study, Li et al. provide novel insights into the role NCF4 in the process of inflammasome assembly and activation. Furthermore, the data presented in the manuscript indicate a potentially crucial role of NCF4 in the process of colorectal cancer formation and progression by controlling immune surveillance.

The authors first identified NCF4 as one of the ASC binding proteins. Indeed, the authors have further shown that NCF4 phosphorylation is crucial for inflammasome activation. Moreover, Ncf4-deficient mice showed a higher tumor burden in two independent murine CRC models. Those results are in line with the analysis of TCGA RNA sequencing data in which the authors observed a reduced survival of patients with low NCF4 expression levels. Overall, the manuscript offers novel and interesting insights into the molecular mechanisms of inflammasome activation. However, the link between those mechanistic aspects on a (sub)cellular level and their functional role in immune surveillance and tumor progression is not fully supported by the data. Furthermore, the analysis of the publicly available human data set is not adequately explained in the manuscript. I detail my concerns below:

We thank the reviewer for noting the novelty of our study.

Major points:

1) The authors use The Cancer Genome Atlas (TCGA) in their analysis in Figure 1. It is unclear how this data set was analyzed - the authors do not mention it in their results section.

We have added the information to the method section. The gene expression data was analyzed by a commercial software available in Xiantaoxueshu (<https://www.xiantaozi.com>), which integrates the raw data collected from The Cancer Genome Atlas database (TCGA).

2) In Fig. 1f, the authors correlate the expression of NCF4 and NCF1 with tumor stages and claim that “higher tumor stages” are associated with lower expression of NCF. However, the authors selected only the “T” (Tumor size) part of the TNM staging system in their analysis. Choosing a more comprehensive staging system for their analysis (and a clinically more important one) would be more interesting: for instance, the UICC tumor stage (which is based on the TNM staging but not on the “T” alone).

We thank the reviewer for this comment. We performed the correlation analysis with TNM staging, but did not find the lower expression of NCF1 or NCF4 being associated with higher tumor stages. We have now removed Fig. 1f used in the previous version.

3) Fig. 1g shows that low NCF4 expression is associated with shorter survival in CRC patients. As mentioned above, it is not clear how this data set was analyzed. Did the authors pool the “COAD” (Colon Adenocarcinoma) and READ (rectal adenocarcinoma) data sets of the TCGA to obtain a true “CRC” cohort?

The reviewer is correct. We pooled the “COAD” (Colon Adenocarcinoma) and READ (rectal adenocarcinoma) data-sets from the TCGA. We added this information to the revised manuscript.

How was the TCGA patient cohort split into “high” and “low” NCF4 expressing patients? Did the authors use a specific cut-off, the median or quartiles?

When performing a fast analysis using the online data analysis platform for the TCGA (Gepia2; <http://gepia2.cancer-pku.cn/#index>; Tang et. al 2017 doi: 10.1093/nar/gkx247), one does not observe a difference in survival when dividing the pooled COAD and READ cohorts by the median NCF4 expression. In general, the authors should explain in detail how they performed their analysis to make it reproducible.

In our analysis, we employed an FPKM (Fragments Per Kilobase of transcript per Million mapped reads) threshold to categorize the TCGA (The Cancer Genome Atlas) patient cohort into groups with high and low gene expression levels. Specifically, patients whose NCF4 gene FPKM values exceeded 2.8 were assigned to the high-expression group. Conversely, those with NCF4 FPKM values at or below 2.8 were allocated to the low-expression group. This information has been added to the revised manuscript.

4) Can the results of the TCGA be confirmed in another cohort of patients? This would be important

Currently, no other cohorts similar to TCGA for CRC patients are available. However, our analysis results are similar to those from website <https://www.proteinatlas.org/>, which are based on TCGA.

5) The authors describe a critical role of NCF4 for the immunosurveillance in CRC. However, this conclusion is not fully supported by the data. The authors link two observations (increased colonic tumor development in Ncf4-deficient mice and a

decreased infiltration of CD8 and NK cells) and assume a causal link between them. A causal role of CD8 or NK cells for the tumor development in *Ncf4*-deficient mice has not been proven in any way. Furthermore, it is worth noting that the analysis of infiltrating immune cells was performed using the AOM/DSS model – in which the degree of inflammation was also different between WT and *Ncf4*-deficient mice and this might well explain differences in the relative amount of distinct immunological populations infiltrating the mucosa. Further mechanistic experiments testing this link are important.

We performed new experiments and injected recombinant IL-18 into *Ncf4*^{-/-} mice treated with AOM-DSS (Extended Data Fig. 7a). Indeed, we found that IL-18 injection rescued the reduced activation of CD8⁺ T cells and NK cells in AOM-DSS-treated *Ncf4*^{-/-} mice, indicating that impaired inflammasome activation in *Ncf4*^{-/-} mice leads to a dysregulated tumor microenvironment. We have added these new data to Extended Data Fig. 8b, c. Extended Data Fig. 8b,c

6) Related to the above point and Fig7h-i: have the authors also analyzed the absolute amount (and not relative percentage) of infiltrating CD8⁺ T cells and NK cells?

We analyzed the total number of the infiltrated CD8⁺ T cells and NK cells and found a reduction in the absolute number of activated CD8⁺ T and NK cells in *Ncf4*^{-/-} mice. The new data have been added to Extended Data Fig. 8a.

Extended Data Fig. 8a

7) The statement that NCF4 regulates the “proliferation” of anti-tumor NK and CD8 T cells (made in the abstract, the results section and in the discussion) is not supported by the data shown in the manuscript. A lower number of NK and CD8 T cells could be related to lower numbers of infiltrating cells (and not their proliferation). Furthermore, and as mentioned above, it is not even clear if the absolute number of those cells is reduced in the tumors of Ncf4-deficient mice.

We followed the reviewer’s advice and changed the wording from “proliferation” to “infiltration” or “recruitment”.

Minor points

1) The wording in the manuscript is often confusing. One of several examples is the statement “NCF4 is a susceptible gene that is significantly associated with Crohn’s disease and colorectal cancer“ (third paragraph in the results section). Did the authors mean “susceptibility gene”? I recommend carefully checking the manuscript for confusing wording.

We have corrected the error and carefully checked the manuscript.

2) The authors do not provide a reference when referring to the TCGA Data set

We have now added a reference and website link

Reference:

1. I. C. Allen *et al.*, The NLRP3 inflammasome functions as a negative regulator of tumorigenesis during colitis-associated cancer. *J Exp Med* **207**, 1045-1056 (2010).
2. M. H. Zaki *et al.*, The NLRP3 inflammasome protects against loss of epithelial integrity and mortality during experimental colitis. *Immunity* **32**, 379-391 (2010).
3. B. Hu *et al.*, Inflammation-induced tumorigenesis in the colon is regulated by caspase-1 and NLRC4. *Proc Natl Acad Sci U S A* **107**, 21635-21640 (2010).
4. M. H. Zaki, P. Vogel, M. Body-Malapel, M. Lamkanfi, T. D. Kanneganti, IL-18 production downstream of the Nlrp3 inflammasome confers protection against colorectal tumor formation. *J Immunol* **185**, 4912-4920 (2010).
5. M. Saleh, G. Trinchieri, Innate immune mechanisms of colitis and colitis-associated colorectal cancer. *Nature Reviews Immunology* **11**, 9-20 (2011).
6. A. H. Chan, K. Schroder, Inflammasome signaling and regulation of interleukin-1 family cytokines. *J Exp Med* **217**, (2020).
7. T. Zhou *et al.*, IL-18BP is a secreted immune checkpoint and barrier to IL-18 immunotherapy. *Nature* **583**, 609-614 (2020).
8. B. Hu *et al.*, Augmentation of Antitumor Immunity by Human and Mouse CAR T Cells Secreting IL-18. *Cell Rep* **20**, 3025-3033 (2017).
9. C. He *et al.*, NCF4 dependent intracellular reactive oxygen species regulate plasma cell formation. *Redox biology* **56**, 102422 (2022).

REVIEWER COMMENTS

Reviewer #3 (Remarks to the Author):

Thanks to the authors for their responses. While I still consider this paper to be interesting, the queries raised by the reviewers were not sufficiently addressed.

All reviewers independently concluded that insufficient mechanistic insight was provided in the original manuscript. Unfortunately little additional insights were provided in the revised MS, i.e. how phosphorylated NCF4 actually leads to inflammasome activation. The authors have responded by inserting figures from the original manuscript into the rebuttal letter (eg. Figs 6 d, e; Fig 6f is a reformatted version of Fig.6g in the original MS) - do they think that the reviewers did not read the first submission?

Reviewer 1 asked whether IL-18 supplementation in Ncf4^{-/-} would have any effect on their AOM-DSS susceptibility, but this was not examined. The authors showed the effect of IL-18 on increasing IFN γ in CD8 and NK cells, which is not surprising.

Two reviewers raised the point that in many cases, representative confocal images were shown, and were not backed up by statistical representation of the overall findings - this was not sufficiently addressed by the authors.

Reviewer #4 (Remarks to the Author):

I appreciate the efforts of Li et al. to improve the manuscript. However, some of my initial concerns have not been adequately addressed. Furthermore, in the rebuttal/reply letter the authors claim to have performed certain corrections – but then many of those are actually not corrected in the manuscript. Thus, I believe that several points still need to be clarified by the authors.

Major points:

1) Initial comment: The authors use The Cancer Genome Atlas (TCGA) in their analysis in

Figure 1. It is unclear how this data set was analyzed - the authors do not mention it in their results section.

Authors response: We have added the information to the method section. The gene expression data was analyzed by a commercial software available in Xiantaoxueshu (<https://www.xiantaozi.com>), which integrates the raw data collected from The Cancer Genome Atlas database (TCGA)

The authors now indeed mention the analysis of the TCGA in their methods section. It remains unclear whether the software used for analysis has been validated and/or published before. If so, the authors should provide this information – right now, the way the analysis was described by the authors is difficult to be reproduced by other researchers. Additionally, the authors do not mention how the statistical test for significance was performed for the survival analysis (they just provide a p value in the figure) – the figure legend only provides information about t-tests performed, which probably does not apply to the analysis of survival data.

2) Initial comment: In Fig. 1f, the authors correlate the expression of NCF4 and NCF1 with tumor stages and claim that “higher tumor stages” are associated with lower expression of NCF. However, the authors selected only the “T” (Tumor size) part of the TNM staging system in their analysis. Choosing a more comprehensive staging system for their analysis (and a clinically more important one) would be more interesting: for instance the UICC tumor stage (which is based on the TNM staging but not on the “T” alone).

Authors response: We thank the reviewer for this comment. We performed the correlation analysis with TNM staging, but did not find the lower expression of NCF1 or NCF4 being associated with higher tumor stages. We have now removed Fig. 1f used in the previous version.

(see figure in rebuttal pdf document)

The authors performed the requested analysis and found that NCF1, NCF2 and NCF4

expression actually does not correlate with tumor stages – consequently they removed the initial figure and the corresponding claims in the manuscript. The point has thus been adequately addressed.

3) Initial comment part A): Fig. 1g shows that low NCF4 expression is associated with shorter survival in CRC patients. As mentioned above, it is not clear how this data set was analyzed. Did the authors pool the “COAD” (Colon Adenocarcinoma) and READ (rectal adenocarcinoma) data sets of the TCGA to obtain a true “CRC” cohort?

Authors response: The reviewer is correct. We pooled the “COAD” (Colon Adenocarcinoma) and READ (rectal adenocarcinoma) data-sets from the TCGA. We added this information to the revised manuscript.

The authors now provide this information in the figure legend of Figure 1. The point has been adequately addressed.

Initial comment part B): How was the TCGA patient cohort split into “high” and “low” NCF4 expressing patients? Did the authors use a specific cut-off, the median or quartiles? When performing a fast analysis using the online data analysis platform for the TCGA (Gepia2; <http://gepia2.cancer-pku.cn/#index>; Tang et. al 2017 doi: 10.1093/nar/gkx247), one does not observe a difference in survival when dividing the pooled COAD and READ cohorts by the median NCF4 expression. In general, the authors should explain in detail how they performed their analysis to make it reproducible.

Authors response: In our analysis, we employed an FPKM (Fragments Per Kilobase of transcript per Million mapped reads) threshold to categorize the TCGA (The Cancer Genome Atlas) patient cohort into groups with high and low gene expression levels. Specifically, patients whose NCF4 gene FPKM values exceeded 2.8 were assigned to the high- expression group. Conversely, those with NCF4 FPKM values at or below 2.8 were allocated to the low-expression group. This information has been added to the revised manuscript.

As mentioned above, the authors now provide a brief description of the TCGA analysis in the methods section. However, it remains unclear, how the threshold of 2.8 was selected to

divide the patients into the high- and low-expression group. Was this value selected a priori (in this case the authors should provide an explanation for why this particular value of 2.8 was selected) or was this value identified using trial and error or machine learning to identify possible cut-off values at which the patient cohorts are statistically significantly different in terms of survival. The reasoning behind the current analysis remains unclear.

4) Initial comment: Can the results of the TCGA be confirmed in another cohort of patients?

Authors response: Currently, no other cohorts similar to TCGA for CRC patients are available. However, our analysis results are similar to those from website <https://www.proteinatlas.org/>, which are based on TCGA.

If the authors do not have access to another patient cohort then the showing the results of the TCGA is sufficient to provide information about a possible clinical relevance.

5) Initial comment: The authors describe a critical role of NCF4 for the immunosurveillance in CRC. However, this conclusion is not fully supported by the data. The authors link two observations (increased colonic tumor development in Ncf4-deficient mice and a decreased infiltration of CD8 and NK cells) and assume a causal link between them. A causal role of CD8 or NK cells for the tumor development in Ncf4-deficient mice has not been proven in any way. Furthermore, it is worth noting that the analysis of infiltrating immune cells was performed using the AOM/DSS model – in which the degree of inflammation was also different between WT and Ncf4-deficient mice and this might well explain differences in the relative amount of distinct immunological populations infiltrating the mucosa.

Authors response: We performed new experiments and injected recombinant IL-18 into Ncf4^{-/-} mice treated with AOM-DSS (Extended Data Fig. 7a). Indeed, we found that IL-18 injection rescued the reduced activation of CD8⁺ T cells and NK cells in AOM-DSS-treated Ncf4^{-/-} mice, indicating that impaired inflammasome activation in Ncf4^{-/-} mice leads to a dysregulated tumor microenvironment. We have added these new data to Extended Data Fig. 8b, c.

My initial comment was focused on the claim that the decreased infiltration of CD8 and NK

cells is causally linked to the increased tumor development in Ncf4-deficient mice. The new data provided by the authors suggested that inflammasome activation can boost the activation of CD8 T cells and NK cells. However, this is per se no experimental evidence that the phenotype observed in Ncf4-deficient mice is indeed dependent on the decreased cellular numbers of CD8+ T cells and NK cells. The data provided shows that it is one of the possible mechanisms, but there is no proof of a causal link based on the data provided. The authors should discuss this point to avoid drawing false conclusions by the reader. Did mice receiving recombinant IL-18 show a phenotype in the development of intestinal tumors?

6) Initial comment: Related to the above point and Fig7h-i: have the authors also analyzed the absolute amount (and not relative percentage) of infiltrating CD8+ T cells and NK cells?

Authors response: We analyzed the total number of the infiltrated CD8+ T cells and NK cells and found a reduction in the absolute number of activated CD8+ T and NK cells in Ncf4^{-/-} mice. The new data have been added to Extended Data Fig. 8a.

This point has been adequately addressed by the authors.

7) Initial comment: The statement that NCF4 regulates the “proliferation” of anti-tumor NK and CD8 T cells (made in the abstract, the results section and in the discussion) is not supported by the data shown in the manuscript. A lower number of NK and CD8 T cells could be related to lower numbers of infiltrating cells (and not their proliferation). Furthermore, and as mentioned above, it is not even clear if the absolute number of those cells is reduced in the tumors of Ncf4-deficient mice.

Authors response: We followed the reviewer’s advice and changed the wording from “proliferation” to “infiltration” or “recruitment”.

Unfortunately, despite the claims of the authors, they have not changed the wording in many parts of the manuscript. For example, in the abstract the authors still write: “This NCF4- coordinated response is necessary to trigger inflammasome-mediated

activation and proliferation of anti-tumor CD8+ T and NK cells and the prevention of colorectal tumorigenesis”.

Similarly, in the discussion the authors state:

“Our single-cell RNA-seq analysis between WT and Ncf4^{-/-} mice revealed critical roles for NCF4 in preventing continuum enterocyte transformation from normal to stem-like and precancerous states by modulating the proliferation and activation of NK and CD8+ T cells within the colon tissue microenvironment. “

At other places of the manuscript, the authors did indeed change the wording, for example to “recruitment”:

“Collectively, single-cell analyses revealed continual cell transformation from normal to precancerous and cancerous states⁴², crucial roles of IFN- γ signaling network in preventing this transformation progress, and NCF4 plays pivotal roles attenuating this transformation by driving inflammasome-dependent activation and recruitment of anti-tumor NK and CD8+ T cells during early stages of cancer development (Extended Data Fig. 8d). “

This correction made by the authors is also not supported by the data – it represents exchanging one of many possible mechanisms by another, both of which are possible but not supported by evidence presented in the manuscript. In my initial comment, I suggested that the statement “NCF4 regulates the proliferation of anti-tumor NK and CD8 T cells” was not supported by the data since a lower number of NK and CD8 T cells could for instance be related to lower numbers of infiltrating cells (and not their proliferation). The authors now simply changed the wording to recruitment. The issue is the same as with “proliferation” – none of those processes were directly studied and thus are not supported by the data. The authors should describe their data in a correct way and not draw conclusions that might confuse the reader. In that case, the authors data show that Ncf4-deficient mice show a lower number of NK and CD8 T cells – whether this is related to proliferation, recruitment, apoptosis or other mechanisms is not known.

Minor points

1) Initial comment: The wording in the manuscript is often confusing. One of several examples is the statement “NCF4 is a susceptible gene that is significantly associated with

Crohn's disease and colorectal cancer" (third paragraph in the results section). Did the authors mean "susceptibility gene"? I recommend carefully checking the manuscript for confusing wording.

Author's response: We have corrected the error and carefully checked the manuscript. This point has been adequately addressed by the authors.

2) Initial comment: The authors do not provide a reference when referring to the TCGA Data set.

Authors response: We have now added a reference and website link

The authors claim to have added a reference to the TCGA and a website link. Unfortunately, this is not true and the TCGA has not been referenced by the authors. The authors only provided a link to the website <https://www.xiantaozi.com> (which refers to a commercial software website according the authors).

Reviewer #5 (Remarks to the Author):

ASC specks should have been quantified in Fig. 6h and f. Aside this minor issue, the authors have done an excellent job in addressing concerns raised by reviewer 1.

REVIEWER COMMENTS

Reviewer #3 (Remarks to the Author):

Thanks to the authors for their responses. While I still consider this paper to be interesting, the queries raised by the reviewers were not sufficiently addressed.

All reviewers independently concluded that insufficient mechanistic insight was provided in the original manuscript. Unfortunately little additional insights were provided in the revised MS, i.e. how phosphorylated NCF4 actually leads to inflammasome activation. The authors have responded by inserting figures from the original manuscript into the rebuttal letter (eg. Figs 6 d, e; Fig 6f is a reformatted version of Fig.6g in the original MS) - do they think that the reviewers did not read the first submission?

Reviewer 1 asked whether IL-18 supplementation in *Ncf4*^{-/-} would have any effect on their AOM-DSS susceptibility, but this was not examined. The authors showed the effect of IL-18 on increasing IFN γ in CD8 and NK cells, which is not surprising.

Two reviewers raised the point that in many cases, representative confocal images were shown, and were not backed up by statistical representation of the overall findings - this was not sufficiently addressed by the authors.

Thanks for your suggestion. We have added the quantification data for most of confocal images including Figure 6h and Extended Data Figure 6f. (Please check the data in response to Reviewer #5)

We performed H&E staining of colon tissues from AOM-DSS treated *Ncf4*^{-/-} mice in the presence or absence of IL-18 administration, and found that IL-18 attenuated the epithelial dysplasia and tumorigenesis of *Ncf4*^{-/-} mice. We have added these new data to Extended Data Figure 8d,e.

Extended Data Figure 8d,e.

Reviewer #4 (Remarks to the Author):

I appreciate the efforts of Li et al. to improve the manuscript. However, some of my initial concerns have not been adequately addressed. Furthermore, in the rebuttal/reply letter the authors claim to have performed certain corrections – but then many of those are actually not corrected in the manuscript. Thus, I believe that several points still need to be clarified by the authors.

Major points:

1) Initial comment: The authors use The Cancer Genome Atlas (TCGA) in their analysis in Figure 1. It is unclear how this data set was analyzed - the authors do not mention it in their results section.

Authors response: We have added the information to the method section. The gene expression data was analyzed by a commercial software available in Xiantaoxueshu (<https://www.xiantaozi.com>), which integrates the raw data collected from The Cancer Genome Atlas database (TCGA)

The authors now indeed mention the analysis of the TCGA in their methods section. It remains unclear whether the software used for analysis has been validated and/or published before. If so, the authors should provide this information – right now, the way the analysis was described by the authors is difficult to be reproduced by other researchers. Additionally, the authors do not mention how the statistical test for significance was performed for the survival analysis (they just provide a p value in the figure) – the figure legend only provides information about t-tests performed, which probably does not apply to the analysis of survival data.

Thanks for your question. Since the contents about how TCGA data set was analyzed were somewhat lengthy, we have added the information about how TCGA data set was analyzed in the method section.

Briefly, we first downloaded expression data and clinical data of Colon and Rectum Adenocarcinomas (TCGA-COAD, TCGA-READ project) from TCGA website (<https://portal.gdc.cancer.gov>). Gene read counts were converted to fragments per kilobase per million (FPKM). Expression comparison between tumor and normal tissue, and in tumor-normal matched pairs was performed using the Wilcoxon test after transforming the FPKMs via $\log_2(\text{FPKM}+1)$. The gene expression values $\log_2(\text{FPKM}+1)$ were then displayed using the scatter and box plot. Survival analysis was performed by using R package survival. We used the Cox proportional hazard model to calculate the hazard ratio (HR), and the 95% confidence interval (CI) was reported. The Kaplan-Meier survival curve was modeled by survfit function. We used the maxstat.test function of R package maxstat to identify best cutting points. This function was used to perform a dichotomy of gene expression and divided the

patients into two groups according to the selected maximum logarithm statistics. The two-sided long-rank test was used to compare Kaplan-Meier survival curves.

We have utilized a well-established commercial platform Xiantaoxueshu (<https://www.xiantaozi.com>) for the visualization or plotting of final processed data.

2) Initial comment: In Fig. 1f, the authors correlate the expression of NCF4 and NCF1 with tumor stages and claim that “higher tumor stages” are associated with lower expression of NCF. However, the authors selected only the “T” (Tumor size) part of the TNM staging system in their analysis. Choosing a more comprehensive staging system for their analysis (and a clinically more important one) would be more interesting: for instance the UICC tumor stage (which is based on the TNM staging but not on the “T” alone).

Authors response: We thank the reviewer for this comment. We performed the correlation analysis with TNM staging, but did not find the lower expression of NCF1 or NCF4 being associated with higher tumor stages. We have now removed Fig. 1f used in the previous version.

(see figure in rebuttal pdf document)

The authors performed the requested analysis and found that NCF1, NCF2 and NCF4 expression actually does not correlate with tumor stages – consequently they removed the initial figure and the corresponding claims in the manuscript. **The point has thus been adequately addressed.**

3) Initial comment part A): Fig. 1g shows that low NCF4 expression is associated with shorter survival in CRC patients. As mentioned above, it is not clear how this data set was analyzed. Did the authors pool the “COAD” (Colon Adenocarcinoma) and READ (rectal adenocarcinoma) data sets of the TCGA to obtain a true “CRC” cohort?

Authors response: The reviewer is correct. We pooled the “COAD” (Colon Adenocarcinoma) and READ (rectal adenocarcinoma) data-sets from the TCGA. We added this information to the revised manuscript.

The authors now provide this information in the figure legend of Figure 1. **The point has been adequately addressed.**

Initial comment part B): How was the TCGA patient cohort split into “high” and “low” NCF4 expressing patients? Did the authors use a specific cut-off, the median or quartiles? When performing a fast analysis using the online data analysis platform for the TCGA (Gepia2; <http://gepia2.cancer-pku.cn/#index>; Tang et. al 2017 doi: 10.1093/nar/gkx247), one does not observe a difference in survival when dividing the pooled COAD and READ cohorts by the median NCF4 expression. In general, the authors should explain in detail how they performed their analysis to make it reproducible.

Authors response: In our analysis, we employed an FPKM (Fragments Per Kilobase of transcript per Million mapped reads) threshold to categorize the TCGA (The Cancer Genome Atlas) patient cohort into groups with high and low gene expression levels. Specifically, patients whose NCF4 gene FPKM values exceeded 2.8 were assigned to the high- expression group. Conversely, those with NCF4 FPKM values at or below 2.8 were allocated to the low-expression group. This information has been added to the revised manuscript.

As mentioned above, the authors now provide a brief description of the TCGA analysis in the methods section. However, it remains unclear, how the threshold of 2.8 was selected to divide the patients into the high- and low-expression group. Was this value selected a priori (in this case the authors should provide an explanation for why this particular value of 2.8 was selected) or was this value identified using trial and error or machine learning to identify possible cut-off values at which the patient cohorts are statistically significantly different in terms of survival. The reasoning behind the current analysis remains unclear.

Thank you for your insightful comments and questions regarding the methodology of our analysis involving the TCGA patient cohort. We apologize for any confusion caused by the lack of detail in our previous submission.

To address your question about the division of the cohort into "high" and "low" NCF4 expressing patients, we did not arbitrarily select a threshold or use a simple median or quartile method. Instead, we employed a statistically robust approach to determine the optimal cut-off point.

We utilized the "maxstat.test" function from the R package "maxstat" to identify the most significant dividing point for NCF4 expression levels. This function operates through repeatedly testing all potential cut-off points to identify the maximum rank statistic, thereby optimizing the dichotomy of gene expression levels(1). This method was widely used in previous publications(2-4). By using this method, we were able to divide the patients into two distinct groups based on the selected maximum logarithm statistics, which provided us with the clearest distinction in terms of survival outcomes. Therefore, the threshold value of 2.8 was determined by the "maxstat.test" function as the point that maximizes the difference in survival between the two groups. We have provided a detailed description of this process in the revised manuscript (see details in question 1).

4) Initial comment: Can the results of the TCGA be confirmed in another cohort of patients?

Authors response: Currently, no other cohorts similar to TCGA for CRC patients are available. However, our analysis results are similar to those from website <https://www.proteinatlas.org/>, which are based on TCGA.

If the authors do not have access to another patient cohort then the showing the results of the TCGA **is sufficient to** provide information about a possible clinical relevance.

5) Initial comment: The authors describe a critical role of NCF4 for the immunosurveillance in CRC. However, this conclusion is not fully supported by the data. The authors link two observations (increased colonic tumor development in *Ncf4*-deficient mice and a decreased infiltration of CD8 and NK cells) and assume a causal link between them. A causal role of CD8 or NK cells for the tumor development in *Ncf4*-deficient mice has not been proven in any way. Furthermore, it is worth noting that the analysis of infiltrating immune cells was performed using the AOM/DSS model – in which the degree of inflammation was also different between WT and *Ncf4*-deficient mice and this might well explain differences in the relative amount of distinct immunological populations infiltrating the mucosa.

Authors response: We performed new experiments and injected recombinant IL-18 into *Ncf4*^{-/-} mice treated with AOM-DSS (Extended Data Fig. 7a). Indeed, we found that IL-18 injection rescued the reduced activation of CD8⁺ T cells and NK cells in AOM-DSS-treated *Ncf4*^{-/-} mice, indicating that impaired inflammasome activation in *Ncf4*^{-/-} mice leads to a dysregulated tumor microenvironment. We have added these new data to Extended Data Fig. 8b, c.

My initial comment was focused on the claim that the decreased infiltration of CD8 and NK cells is causally linked to the increased tumor development in *Ncf4*-deficient mice. The new data provided by the authors suggested that inflammasome activation can boost the activation of CD8 T cells and NK cells. However, this is per se no experimental evidence that the phenotype observed in *Ncf4*-deficient mice is indeed dependent on the decreased cellular numbers of CD8⁺ T cells and NK cells. **The data provided shows that it is one of the possible mechanisms**, but there is no proof of a causal link based on the data provided. The authors should discuss this point to avoid drawing false conclusions by the reader. Did mice receiving recombinant IL-18 show a phenotype in the development of intestinal tumors?

6) Initial comment: Related to the above point and Fig7h-i: have the authors also analyzed the absolute amount (and not relative percentage) of infiltrating CD8⁺ T cells and NK cells?

Authors response: We analyzed the total number of the infiltrated CD8⁺ T cells and NK cells and found a reduction in the absolute number of activated CD8⁺ T and NK cells in *Ncf4*^{-/-} mice. The new data have been added to Extended Data Fig. 8a.

This point has been adequately addressed by the authors.

7) Initial comment: The statement that NCF4 regulates the “proliferation” of anti-tumor NK and CD8 T cells (made in the abstract, the results section and in the discussion) is not supported by the data shown in the manuscript. A lower number of NK and CD8 T cells could be related to lower numbers of infiltrating cells (and not their proliferation).

Furthermore, and as mentioned above, it is not even clear if the absolute number of those cells is reduced in the tumors of *Ncf4*-deficient mice.

Authors response: We followed the reviewer's advice and changed the wording from "proliferation" to "infiltration" or "recruitment".

Unfortunately, despite the claims of the authors, they have not changed the wording in many parts of the manuscript. For example, in the abstract the authors still write:

"This NCF4- coordinated response is necessary to trigger inflammasome-mediated activation and proliferation of anti-tumor CD8+ T and NK cells and the prevention of colorectal tumorigenesis".

Similarly, in the discussion the authors state:

"Our single-cell RNA-seq analysis between WT and *Ncf4*^{-/-} mice revealed critical roles for NCF4 in preventing continuum enterocyte transformation from normal to stem-like and precancerous states by modulating the proliferation and activation of NK and CD8+ T cells within the colon tissue microenvironment. "

At other places of the manuscript, the authors did indeed change the wording, for example to "recruitment":

"Collectively, single-cell analyses revealed continual cell transformation from normal to precancerous and cancerous states⁴², crucial roles of IFN- signaling network in preventing this transformation progress, and NCF4 plays pivotal roles attenuating this transformation by driving inflammasome-dependent activation and recruitment of anti-tumor NK and CD8+ T cells during early stages of cancer development (Extended Data Fig. 8d). "

This correction made by the authors is also not supported by the data – it represents exchanging one of many possible mechanisms by another, both of which are possible but not supported by evidence presented in the manuscript. In my initial comment, I suggested that the statement "NCF4 regulates the proliferation of anti-tumor NK and CD8 T cells" was not supported by the data since a lower number of NK and CD8 T cells could for instance be related to lower numbers of infiltrating cells (and not their proliferation). The authors now simply changed the wording to recruitment. The issue is the same as with "proliferation" – none of those processes were directly studied and thus are not supported by the data. The authors should describe their data in a correct way and not draw conclusions that might confuse the reader. In that case, the authors data show that *Ncf4*-deficient mice show a lower number of NK and CD8 T cells – whether this is related to proliferation, recruitment, apoptosis or other mechanisms is not known.

Thanks for your suggestion. We have changed the word "proliferation" to "increased population" or "population" to correctly describe the results and conclusion.

We performed H&E staining of colon tissues from AOM-DSS treated *Ncf4*^{-/-} mice in the presence or absence of IL-18 administration, and found that IL-18 attenuated the

epithelial dysplasia and tumorigenesis of *Ncf4*^{-/-} mice. We have added these new data to Extended Data Figure 8d,e.

Extended Data Figure 8d,e.

Minor points

1) Initial comment: The wording in the manuscript is often confusing. One of several examples is the statement “NCF4 is a susceptible gene that is significantly associated with Crohn’s disease and colorectal cancer” (third paragraph in the results section). Did the authors mean “susceptibility gene”? I recommend carefully checking the manuscript for confusing wording.

Author’s response: We have corrected the error and carefully checked the manuscript.

This point has been adequately addressed by the authors.

2) Initial comment: The authors do not provide a reference when referring to the TCGA Data set.

Authors response: We have now added a reference and website link

The authors claim to have added a reference to the TCGA and a website link. Unfortunately, this is not true and the TCGA has not been referenced by the authors. The authors only provided a link to the website <https://www.xiantaozi.com> (which refers to a commercial software website according to the authors).

We have added the TCGA website <https://portal.gdc.cancer.gov>.

Reviewer #5 (Remarks to the Author):

ASC specks should have been quantified in Fig. 6h and f. Aside this minor issue, the authors have done an excellent job in addressing concerns raised by reviewer 1.

Thanks a lot for your comments. We have added the quantification data to **Extended Data Figure 6g**.

Reference:

1. T. Hothorn, A. Zeileis, Generalized Maximally Selected Statistics. *Biometrics* **64**, 1263-1269 (2008).
2. T. Duhon *et al.*, Co-expression of CD39 and CD103 identifies tumor-reactive CD8 T cells in human solid tumors. *Nat Commun* **9**, 2724 (2018).
3. M. Zapatka *et al.*, The landscape of viral associations in human cancers. *Nature Genetics* **52**, 320-330 (2020).
4. J. A. Seoane, J. G. Kirkland, J. L. Caswell-Jin, G. R. Crabtree, C. Curtis, Chromatin regulators mediate anthracycline sensitivity in breast cancer. *Nat Med* **25**, 1721-1727 (2019).

REVIEWERS' COMMENTS

Reviewer #4 (Remarks to the Author):

In the previous revision round, the authors have incompletely addressed some of the concerns initially raised by me. However, Li et al. have now indeed improved their manuscript more thoroughly. Overall, my points have now been adequately addressed.

REVIEWERS' COMMENTS

Reviewer #4 (Remarks to the Author):

In the previous revision round, the authors have incompletely addressed some of the concerns initially raised by me. However, Li et al. have now indeed improved their manuscript more thoroughly. Overall, my points have now been adequately addressed.

Response:

Thanks a lot!